# The hepatitis E virus capsid protein ORF2 counteracts cell-intrinsic antiviral responses to enable persistent replication in cell culture

Ann-Kathrin Mehnert[1,2☯], Sebastian Stegmaier[3,4☯], Carlos Ramirez Alvarez[5☯], Elif Toprak[1,2], Vladimir Gonçalves Magalhães[4], Carla Siebenkotten[1], Jungen Hu[1,2], Ana Luisa Costa[5], Daniel Kirrmaier[6,7], Michael Knop[6,7], Xianfang Wu[8], Thibault Tubiana[9], Carl Herrmann[10*], Marco Binder[4], Viet Loan Dao Thi[1,10*]

1 Schaller Research Group, Department of Infectious Diseases, Virology, Heidelberg University, Medical Faculty Heidelberg, Heidelberg, Germany, 2 Heidelberg Biosciences International Graduate School (HBIGS), Heidelberg, Germany, 3 Faculty of Biosciences, Heidelberg University, Heidelberg, Germany, 4 Research Group "Dynamics of Early Viral Infection and the Innate Antiviral Response", Division Virus-Associated Carcinogenesis (D430), German Cancer Research Center (DKFZ), Heidelberg, Germany, 5 Institute for Pharmacy and Molecular Biotechnology, Heidelberg University & BioQuant, Heidelberg, Germany, 6 Center for Molecular Biology of Heidelberg University (ZMBH), Heidelberg, Germany, 7 Division Cell Morphogenesis and Signal Transduction (A260), German Cancer Research Center (DKFZ), Heidelberg, Germany, 8 Infection Biology Program and Department of Cancer Biology, Lerner Research Institute, Cleveland Clinic Foundation, Cleveland, Ohio, United States of America, 9 Institute for Integrative Biology of the Cell (I2BC), Université Paris-Saclay, CEA, CNRS, Gif-sur-Yvette, France, 10 German Center for Infection Research (DZIF), Partner Site Heidelberg, Heidelberg, Germany

☯ These authors contributed equally to this work.
* Carl.Herrmann@uni-heidelberg.de (CH); VietLoan.DaoThi@med.uni-heidelberg.de (VLDT)

## Abstract

Hepatitis E virus (HEV) is a significant human pathogen causing both acute and chronic infections worldwide. The cell-intrinsic antiviral response serves as the initial defense against viruses and has been shown to be activated upon HEV infection. HEV can replicate in the presence of this response, but the underlying mechanisms remain poorly understood. Here, we investigated the roles of the structural proteins ORF2 and ORF3 in the cell-intrinsic antiviral response to HEV infection. Mechanistically, we validated that ectopic ORF2, but not ORF3, interfered with antiviral and inflammatory signaling downstream of pattern recognition receptors, in part through interaction with the central adaptor protein TANK binding kinase 1. In the full-length viral context, ORF2 contributed to a reduced antiviral response and consequently, more efficient viral replication. In addition, we discovered a protective mechanism mediated by ORF2 that shielded viral replication from antiviral effectors. Using single-cell RNA-sequencing, we confirmed that the presence of ORF2 in infected cells dampened antiviral responses in both actively infected cells and bystanders. As a consequence, we found that early in the infection process, the progression of authentic HEV infection relied on the presence of ORF2, facilitating a balance between viral replication and the antiviral response. Altogether, our findings shed

**Data availability statement:** The scRNA-seq data is accessible under the GEO accession number GSE288400. All scripts used and models produced in this study with AlphaFold 2.3 are available on Zenodo (https://doi.org/10.5281/zenodo.14751497).

**Funding:** A.M., S.S., C.R., C.H., M.B., and V.L.D.T were supported by grants from the Deutsche Forschungsgemeinschaft (DFG, German Research Foundation) – Projektnummer – 272983813 SFB/TRR 179. V.L.D.T. was additionally supported by the Chica and Heinz Schaller Foundation, DFG grant DA1640/3-1, and TTU Hepatitis Project 05.823. J.H. was supported by a fellowship from the China Scholarship Council. For the publication fee, we acknowledge financial support by Heidelberg University and TRR 179.The funders had no role in study design, data collection and analysis, decision to publish, or preparation of the manuscript.

**Competing interests:** The authors have declared that no competing interests exist.

new light on the multifaceted role of ORF2 in the HEV life cycle and improve our understanding of the determinants that contribute to persistent HEV replication in cell culture.

## Author summary

Hepatitis E virus (HEV) is an important yet often underestimated pathogen. Depending on the genotype, HEV infections can progress to chronicity, but the underlying mechanisms remain poorly understood. To gain insight into potential determinants, we investigated how HEV evades the cell-intrinsic antiviral response. We discovered that the HEV capsid protein ORF2 is crucial in limiting this response by interfering with antiviral signaling pathways and shielding viral replication from immune effectors. This balance between viral replication and the antiviral response contributes to persistent HEV infection in cell culture. Our findings reveal a new role for the HEV capsid protein in the viral life cycle and highlight it as an important target for novel therapeutic approaches.

## Introduction

Hepatitis E virus (HEV) is one of the major causes of acute hepatitis across the globe, affecting an estimated 20 million people every year [1,2]. As part of the *Hepeviridae* family, the *Paslahepevirus balayani* genus comprises eight genotypes of which HEV-1 to -4, and more recently HEV-7, have been associated with human infections. HEV-1 and -2 are transmitted fecal-orally and are restricted to humans. In contrast, HEV-3 and -4 spread to humans zoonotically through consumption of meat products from domestic pigs or wild boar. Although HEV-1 and -2 infections are usually self-limiting, they are associated with high mortality in pregnant women. Further, HEV-3 and -4 infections of immunocompromised individuals can result in chronicity and development of liver fibrosis and cirrhosis.

HEV is ingested and excreted as a naked virus but circulates in the bloodstream wrapped in a host membrane-derived quasi-envelope, which is acquired during budding from cells [3]. HEV has a positive-sense, single-stranded RNA genome of ~7.2 kilobases, encompassing three open reading frames (ORFs) that give rise to three viral proteins [2]. ORF1 contains the domains involved in viral replication, such as the RNA-dependent RNA polymerase (RdRp). ORF3 is a small phosphoprotein that mediates secretion of viral progeny [2].

Three distinct isoforms of ORF2 have been described in HEV-infected cells [4–7]. Owing to its N-terminal signal peptide, ORF2 can be secreted along the secretory pathway where it gives rise to the glycosylated (ORF2g) and cleaved (ORF2c) isoforms. ORF2g is secreted as a dimer [5,6] and serves as an immunological decoy *in vivo* [5]. It has been postulated that positively charged residues within an arginine-rich motif (ARM; five arginine residues: RRRGRR) downstream of the signal

peptide assist ORF2 in retaining a cytosolic orientation at the endoplasmic reticulum membrane [4]. The intracellular ORF2 isoform (ORF2i) assembles into infectious progenies. ORF2i is mainly located in the cytosol, but can translocate to the nucleus, and is likely involved in many virus-host interactions [4,7]. A different study highlighted the importance of two ORF2 start codons. The first start codon gives rise to the secreted ORF2g, while the second gives rise to ORF2i with intracellular localization [5]. Although it remains unclear whether differential usage of these start codons is relevant in HEV infection, they can be used to abrogate expression of individual or all ORF2 isoforms.

Cell-intrinsic defense strategies are initiated by recognition of pathogen-associated molecular patterns (PAMPs) through pattern recognition receptors (PRRs). In epithelial cells, double-stranded (ds)RNA, the replication intermediate of RNA viruses, is detected by the family of retinoic acid-inducible gene I (RIG-I)-like receptors (RLRs) in the cytosol, including RIG-I and melanoma differentiation-associated protein 5 (MDA5), which are additionally regulated by laboratory of genetics and physiology 2 (LGP2) [8]. In the endosome, dsRNA is recognized by Toll-like receptor 3 (TLR3) [9]. Different adaptor proteins mediate downstream signaling: Toll/interleukin-1 receptor (TIR) domain-containing adaptor-inducing interferon-β (TRIF) is recruited by TLR3, while RIG-I and MDA5 induce polymerization of mitochondrial antiviral signaling protein (MAVS) [10]. Both result in activation of TANK binding kinase 1 (TBK1) and the inhibitor of nuclear factor-κB (IκB) kinase (IKK) complex. TBK1 phosphorylates interferon regulatory factor 3 (IRF3), which translocates into the nucleus and induces expression of type I and type III interferons (IFNs) and IFN-stimulated genes (ISGs) [10]. Autocrine and paracrine IFN signaling through Janus kinase/signal transducers and activators of transcription (JAK/STAT) result in expression of hundreds of ISGs with antiviral functions [11]. Activation of the IKK complex leads to nuclear translocation of nuclear factor-κB (NF-κB) and expression of inflammatory cytokines [12].

Despite growing interest and advances in recent years, the cell-intrinsic antiviral response to HEV infection remains insufficiently described. HEV-infected chimpanzees had a similar but weaker ISG response compared to hepatitis C virus (HCV) infection, likely owing to the lower HEV viremia [13]. Infection of different hepatocellular models with HEV induced robust type III IFN responses, which were dependent on viral replication [14,15]. These responses appeared to be persistent but were unable to eliminate viral replication within the observed time frames [14,15]. Blunting of IFN induction did not result in enhanced viral replication, suggesting that HEV has developed mechanisms to persistently replicate in the presence of a sustained antiviral response [15]. Corroborating this, several studies have shown that HEV replication is relatively resistant to exogenous IFN treatment, as compared to, for example, HCV [14,16–18]. Yin and colleagues proposed that this is the result of an internal IFN refractoriness due to persistent activation of JAK/STAT1 signaling and retention of phosphorylated STAT1 in the cytosol, rather than a direct viral antagonism [14].

Nonetheless, all HEV proteins have been previously suggested to antagonize the cell-intrinsic antiviral response [19]. Within ORF1, the X domain, the putative papain-like cysteine protease (PCP) domain, and the methyltransferase (MeT) domain were shown to interfere with type I IFN induction and NF-κB-mediated signaling [20–23]. A combined MeT-Y-PCP polyprotein was further found to interfere with the JAK/STAT pathway [24]. The ORF2 protein was reported to interact with TBK1 [25,26] and hinder activation of NF-κB [4,27]. Studies of potential ORF3-mediated antagonisms have been controversial: while several reports suggested that ORF3 interferes with type I IFN production, NF-κB-, and IFN-mediated signaling [28–30], one study showed that ORF3 could enhance IFN induction [31]. Most investigations relied exclusively on overexpression of individual HEV proteins or subdomains of ORF1, and consequently, the physiological relevance of each antagonism for authentic HEV replication remains elusive.

In the present study, we sought to clarify the role of HEV ORF2 and ORF3 proteins in antagonizing antiviral responses and their contribution to the progression of HEV infection. To address this, we made use of viral mutants and assessed differences in IFN and ISG responses upon authentic infection of different hepatocellular systems in bulk and at single-cell resolution. We identified a replication-limiting bottleneck mediated by the antiviral response early in infection that is overcome by the capsid protein ORF2, but not ORF3, allowing for the establishment of persistent viral replication.

## Results

### The HEV ORF2 protein antagonizes both antiviral and inflammatory signaling pathways and interacts with TBK1

Several independent studies reported that HEV ORF2 and ORF3 antagonize different steps of the cell-intrinsic antiviral response, however, with different conclusions [25–30]. Therefore, we aimed to mechanistically dissect at which steps ORF2 and ORF3 interfere with antiviral and inflammatory signaling. Importantly, we clearly separated between IFN induction and IFN signaling and systematically compared ORF2 and ORF3 side-by-side.

We first focused on sensing and signaling by PRRs. For this mechanistic investigation, we made use of A549-derived cell lines, which harbor a double knockout of RIG-I and MDA5 and show negligible TLR3 expression [32]. In addition to reconstitution with weak transgene expression of either MDA5 [33], RIG-I [33], or TLR3, we ectopically expressed HEV ORF2, HEV ORF3, or GFP in these cells. We confirmed expression of ORF2 and ORF3 proteins by Western blot (WB) analysis (S1A and S1B Fig).

In GFP-expressing cells, stimulation of the respective PRR with Mengo-Zn virus, Sendai virus (SeV), and poly(I:C) supernatant feeding resulted in the induction of *IFNB1* expression (Fig 1A–C). Ectopic expression of ORF3 did not significantly alter *IFNB1* expression downstream of RIG-I and MDA5 (Fig 1A and 1B) but resulted in a minor enhancement

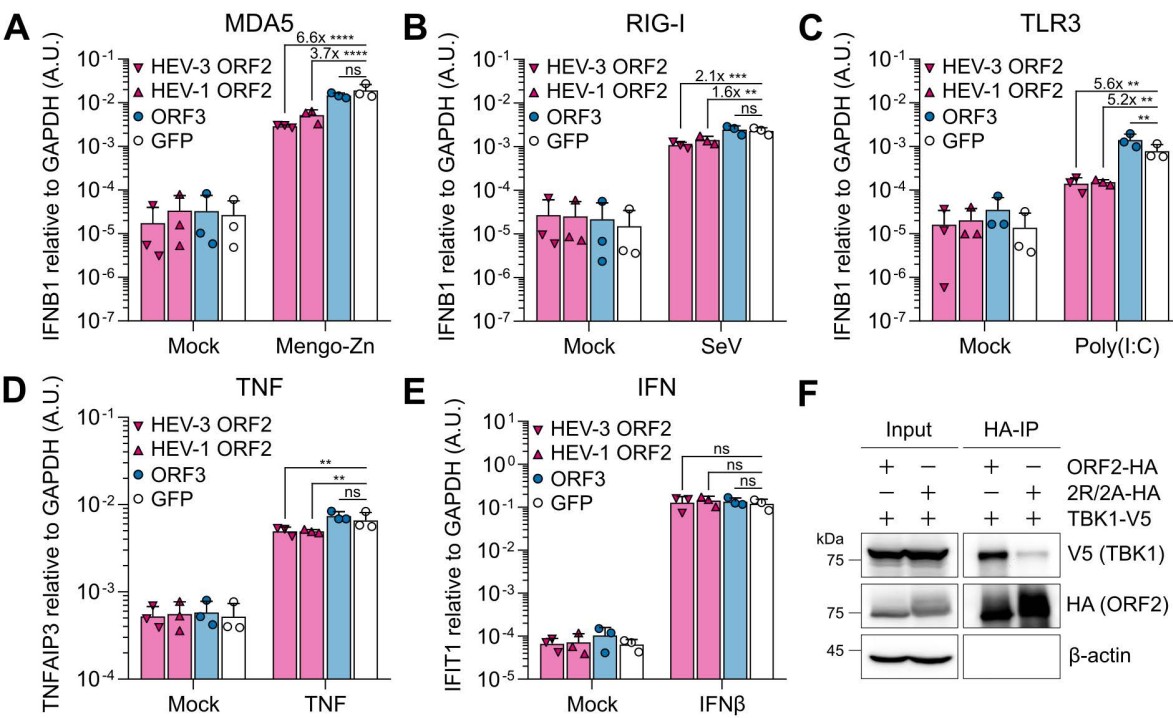

**Fig 1. The HEV ORF2 protein antagonizes both antiviral and inflammatory signaling pathways and interacts with TBK1.** (A) A549 cells harboring knockouts of the PRRs RIG-I and MDA5 and ectopically expressing a single PRR (MDA5, RIG-I, or TLR3) together with either HEV-3 ORF2, HEV-1 ORF2, ORF3, or GFP were challenged with either Mengo-Zn virus at MOI 1 for 24 h, (B) Sendai virus (SeV) at MOI 0.75 for 4 h, or (C) 50 μg/mL poly(I:C) supernatant feeding for 24 h, as indicated. (D) A549-derived cells ectopically expressing MDA5 were stimulated with 10 ng/mL TNF or (E) 200 IU/mL IFNβ for 8 h. At respective time points, (A–C) *IFNB1*, (D) *TNFAIP3*, and (E) *IFIT1* expression were analyzed by RT-qPCR, each relative to the housekeeping gene *GAPDH* using the $2^{-\Delta Ct}$ method. Numbers indicate fold reductions compared to GFP. Data show mean ± SD of n = 3 independent biological experiments. Statistical analysis was performed using two-way ANOVA. **: p < 0.01; ***: p < 0.001; ****: p < 0.0001; ns, non-significant. A.U., arbitrary units. (F) HEK293T cells were co-transfected with ORF2-HA or ORF2-2R/2A-HA and TBK1-V5 and lysed 24 h post-transfection. Anti-HA co-IP and WB analysis for TBK1 (anti-V5 staining), ORF2 (anti-HA staining), and β-actin were performed. Shown is a representative blot of n = 3 independent biological experiments.

downstream of TLR3 compared to the GFP control (Fig 1C). In cells expressing ectopic ORF2, *IFNB1* induction was significantly reduced up to 6.6-fold downstream of all PRRs compared to GFP-expressing cells (Fig 1A–C). Not only ORF2 from a chronic HEV genotype (HEV-3) but also from an acute genotype (HEV-1) dampened this response (Fig 1A–C). By ELISA, we confirmed that ectopic ORF2 expression also significantly reduced IFNβ protein secretion to a similar level as *IFNB1* mRNA (S1C–E Fig). While the presence of ectopic ORF2 dampened the strength of *IFNB1* induction, it did not alter its dynamics, which we tested by time-resolved RT-qPCR analysis of MDA5-expressing cells electroporated with poly(I:C) (S2A and S2D Fig).

We further investigated the impact of ORF2 expression on inflammatory and IFN-dependent signaling. Upon TNF stimulation, the presence of ectopic ORF2, but not ORF3, reduced the induction of the inflammatory cytokine *TNFAIP3* significantly (Fig 1D), thus interfering with NF-κB-dependent signaling. We further confirmed that ORF2 interferes with NF-κB-mediated signal transduction downstream of MDA5 through poly(I:C) electroporation (EPO). Expression of *TNFAIP3* and *IL6* was generally weaker compared to the GFP control, whereas the kinetics remained unaffected (S2B and S2C and S2E and S2F Fig). In contrast, we did not observe significant differences in expression of the ISG *IFIT1* upon stimulation with IFNβ in the presence of either ectopic ORF2 or ORF3, suggesting that neither interfered with JAK/STAT signaling (Fig 1E).

Our results demonstrated that ORF2 interferes with the sensing pathway upstream of IFN induction as well as with NF-κB-dependent inflammatory cytokine induction, similar to previous literature [25–27]. ORF3 did not affect antiviral and inflammatory responses as strongly as proposed previously [28–31]. Importantly, IFN-mediated signaling was not affected by either of the viral proteins. As NF-κB-dependent cytokine induction is particularly important for the crosstalk with immune cells *in vivo*, we continued to solely focus on the effects of ORF2 on the antiviral IFN/ISG response.

It was previously suggested that ORF2 interacts with the critical adaptor molecule TBK1, likely through the ARM [26], but a direct interaction through this motif was not demonstrated. Here, we confirmed the interaction of HEV-3 ORF2 and TBK1 by co-immunoprecipitation (co-IP) (Fig 1F). We further found that mutation of the last two arginine residues of the ARM to alanine (2R/2A) decreased the interaction of TBK1 with ORF2 (Fig 1F). However, this particular mutation also abrogates nuclear translocation of ORF2i and increases ORF2 glycosylation and secretion along the secretory pathway [4]. The pool of cytosolic ORF2i is decreased [4] and consequently, the likelihood of ORF2 interacting with TBK1 is reduced. In agreement with this, the ORF2-2R/2A protein appeared with a light smear in our WB analysis as compared to wild type (WT) ORF2, suggesting the presence of post-translational modifications, such as glycosylation (Fig 1F).

In an effort to determine other potential interaction motifs between ORF2 and TBK1, we made use of AlphaFold. Repeated modeling identified the last three residues of the LGSAWRD motif (spanning amino acids 83–89) at the N-terminus of ORF2 as the only promising, putative interaction motif (S3A and S3C–E Fig). However, mutation of these residues to triple alanine did not result in impaired interaction between ORF2 and TBK1, suggesting that this predicted motif is not essential (S3B Fig).

Altogether, we systematically validated side-by-side that ORF2 but not ORF3 counteracts IFN induction downstream of all relevant PRRs and additionally interferes with NF-κB-mediated signaling. The antagonism of the antiviral response by ORF2 is mediated at least in part by interaction with TBK1, although the precise interaction motif remains elusive.

## The capsid protein ORF2, but not the ORF3 protein, is pivotal for efficient HEV replication in the presence of an antiviral response

HEV replication in hepatocytes persists in the presence of a sustained cell-intrinsic antiviral response [14,15]. To study the extent to which HEV replication is dependent on ORF2-mediated escape of the antiviral response, we generated a ΔORF2 and a control ΔORF3 mutant of the HEV-3 Kernow-C1/p6 WT strain (Fig 2A). To obtain ΔORF2, we mutated the two start codons of *ORF2* to abrogate protein expression of all ORF2 isoforms without affecting correct translation of the overlapping *ORF3*, as suggested by Yin *et al.* [5]. We employed EPO of *in vitro* transcribed (IVT) HEV RNA to bypass virus entry,

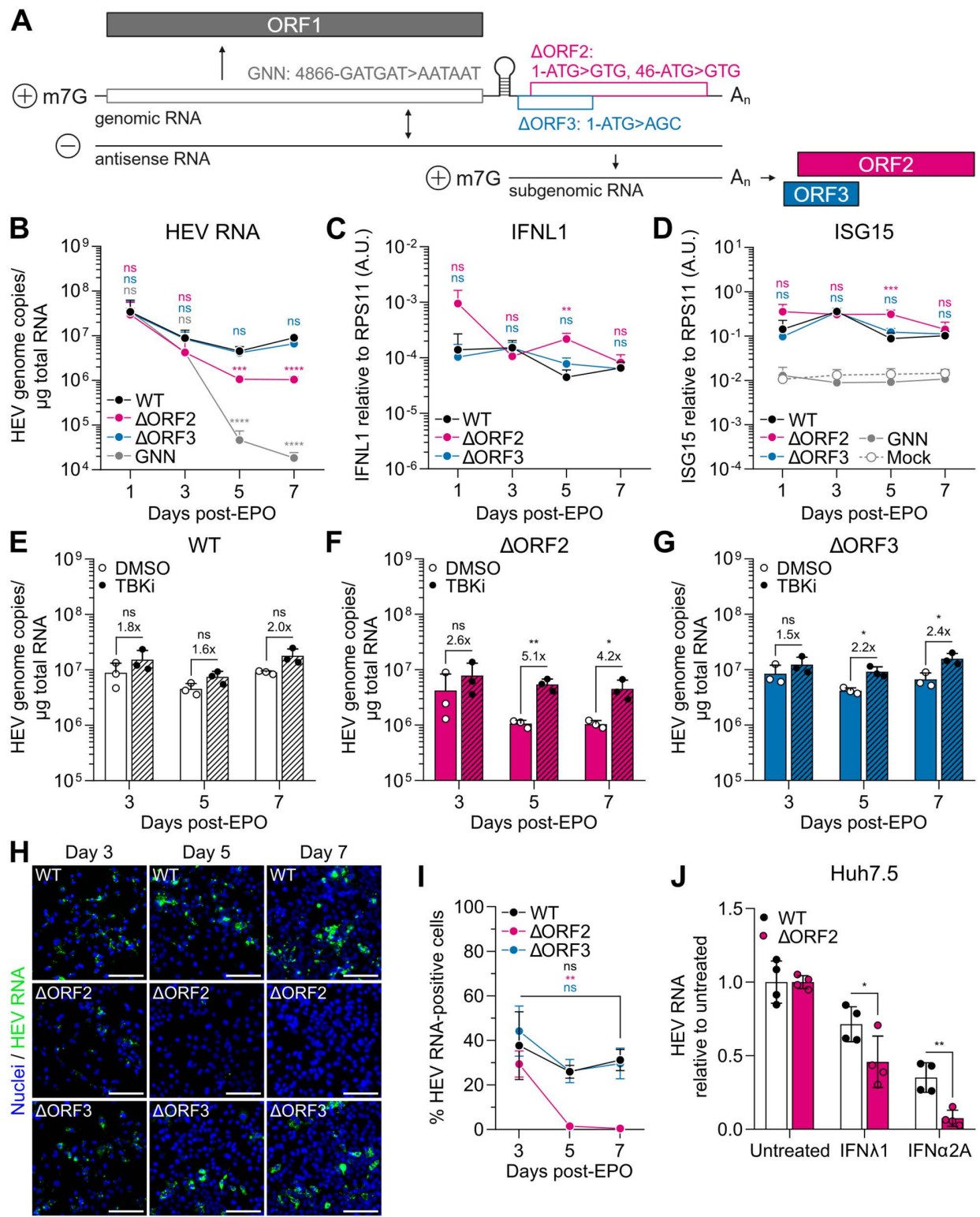

**Fig 2. The capsid protein ORF2, but not the ORF3 protein, is pivotal for efficient HEV replication in the presence of an antiviral response.**
(A) Schematic representation of the HEV genome and its replication intermediates. Nucleic acid positions of GNN and start codon mutations of ΔORF2 and ΔORF3 mutants are indicated. m7G, 7-methylguanosine cap; A$_n$, poly(A)-tail. Created in BioRender. Dao Thi, V. (2025) https://BioRender.com/

j85i934 (B) HepG2/C3A cells were electroporated with IVT HEV WT, ΔORF2, ΔORF3, or replication-incompetent GNN RNA and grown in the presence of 0.06% DMSO 48 h prior to harvesting. Cell lysates were analyzed by RT-qPCR at indicated time points post-EPO for HEV RNA genome copies and (C) *IFNL1* and (D) *ISG15* expression relative to the housekeeping gene *RPS11* using the $2^{-\Delta Ct}$ method. *IFNL1* expression was undetectable in mock and GNN samples. Data show mean ± SD of n = 3 independent biological experiments. Statistical analysis over WT was performed using one-way ANOVA of each time point independently and is indicated above the respective time points in the corresponding color. **: p < 0.01; ***: p < 0.001; ****: p < 0.0001; ns, non-significant. A.U., arbitrary units. (E–G) Electroporated HepG2/C3A cells from (B) were treated with 0.06% DMSO as vehicle control or with 6 μM of the TBK1 inhibitor BX795 (TBKi) 48 h prior to harvesting. HEV RNA was quantified by RT-qPCR and fold changes over DMSO treatment are shown above the respective graphs. Data show mean ± SD of n = 3 independent biological experiments. Statistical analysis was performed using unpaired two-tailed Student's t-test of each time point independently. *: p < 0.05; **: p < 0.01; ns, non-significant. (H) Electroporated HepG2/C3A cells from (B) were stained for HEV RNA using RNA-FISH at indicated days post-EPO. Exemplary images of n = 3 independent biological experiments. Scale bar, 100 μm. (I) The percentages of electroporated HepG2/C3A cells positive for HEV RNA, stained by RNA-FISH in (H), were quantified with ilastik and CellProfiler. Data show mean ± SEM of n = 3 independent biological experiments with 5 randomly selected fields of view quantified per condition of each independent experiment. Statistical analysis was performed using unpaired two-tailed Student's t-test of each condition independently, indicated in the corresponding color. **: p < 0.01; ns, non-significant. (J) Huh7.5 cells were electroporated with IVT HEV WT or ΔORF2 RNA and treated with 10 ng/mL IFNλ1 or 10,000 IU/mL IFNα2A from day 4 to day 7 post-EPO. IFNs were replenished every 24 h. HEV RNA was quantified by RT-qPCR on day 7 post-EPO and normalized over the respective untreated condition. Data show mean ± SD of n = 4 biological repeats from two independent biological experiments. Statistical analysis was performed using two-way ANOVA. *: p < 0.05; **: p < 0.01.

which enabled us to directly compare viral replication and cell-intrinsic antiviral responses of WT and mutated HEV over time, under the same conditions.

Upon EPO into HepG2/C3A cells [14,34], we observed significantly reduced replication of the ΔORF2 mutant compared to WT and the ΔORF3 mutant on days 5 and 7 post-EPO (Fig 2B). Of note, at least part of the basal HEV RNA detected up to day 3 post-EPO resulted from high levels of incoming, electroporated RNA, as revealed by comparison with a replication-incompetent GNN mutant of the RdRp (Fig 2B). *IFNL1* (Fig 2C) and *ISG15* induction (Fig 2D) upon HEV WT, ΔORF3, but also ΔORF2 replication, were similar on days 1 and 3 post-EPO. However, we observed significantly stronger expression of both antiviral response genes for ΔORF2 on day 5 post-EPO (Fig 2C and 2D), coinciding with the decrease in ΔORF2 viral RNA (Fig 2B). While induction of *IFNL1* and *ISG15* was not significantly different from WT and ΔORF3 on day 7 post-EPO (Fig 2C and 2D), the 8.6-fold lower level of ΔORF2 compared to WT RNA at this time point (Fig 2B) indicated a relatively stronger antiviral response induction per ΔORF2 genome, which we highlighted by normalization over HEV RNA in S4A and S4B Fig. Importantly, we did not observe induction of *IFNL1* (not detectable) or *ISG15* expression (Fig 2D) upon EPO with the replication-incompetent GNN mutant. As ORF3 is a critical protein responsible for secretion of viral progeny [2] and the ΔORF3 mutant replicated at WT levels in our experimental setup (Fig 2B), we assumed that potential effects of secondary virus spread were negligible in our system. In addition, newly produced virus particles released into the supernatant are quasi-enveloped and, therefore, not highly infectious [35].

We further validated the stronger IFN response induced by the ΔORF2 mutant on days 5 and 7 post-EPO by measuring secreted IFN proteins in the cell culture supernatants and analyzing STAT1 phosphorylation by WB (S5 Fig). We observed enhanced secretion of IFNλ1 protein for ΔORF2 on all days post-EPO, and the differences became significant on days 5 and 7 (S5A Fig). This enhanced IFN secretion further resulted in elevated levels of phosphorylated STAT1, but not basal STAT1, on days 5 and 7 post-EPO with ΔORF2, compared to WT (S5B Fig). Collectively, we concluded that the lack of the TBK1 inhibition mediated by ORF2 resulted in the stronger induction of an antiviral response in the absence of ORF2.

We also tested whether ORF2 is critical for replication of the non-adapted HEV-3 83-2-27 strain [36], which is devoid of any insertions. EPO of HepG2/C3A cells with the derived ΔORF2 mutant showed lower replication levels compared to WT on day 7 post-EPO (S6A Fig). In contrast, stronger *ISG15* expression was induced on all days post-EPO compared to WT, which was significant on day 3 (S6B Fig). The differences in the kinetics of the ΔORF2 phenotype compared to the Kernow-C1/p6 strain might be a result of the overall lower and possibly slower replication of the 83-2-27 strain.

We hypothesized that the antiviral response induced in the absence of ORF2 was strong enough to cause, at least in part, the observed decline in viral replication. In order to address this, we blunted IFN induction by treating electroporated cells with the TBK1 inhibitor BX795, which had minimal effects on cell viability (S4C–E Fig). The treatment reduced viral replication-induced *ISG15* expression to similar levels for WT, ΔORF2, and ΔORF3 (S4F–H Fig). As expected, HEV WT replication was not significantly enhanced by inhibition of the antiviral response (Fig 2E), while ΔORF3 replication only showed a minor increase (Fig 2G). In contrast, the BX795 treatment resulted in a significant and stronger increase of ΔORF2 replication, especially on days 5 and 7 post-EPO (Fig 2F). This suggested that ΔORF2 replication, in addition to inducing a stronger antiviral response, is also more susceptible to the effects of the induced ISGs.

To study the reduction of ΔORF2 RNA over time at the single-cell level, we used RNA-fluorescence *in situ* hybridization (RNA-FISH) (Fig 2H), which allows the detection of single HEV genomes [37]. This method enabled us to quantify the percentage of HEV-infected cells over time, independent of viral antigen expression. We observed an increase in HEV RNA signal per cell on days 5 and 7 compared to day 3 post-EPO for WT and ΔORF3, indicating an increase in viral replication (Fig 2H). However, the overall percentage of HEV RNA-positive cells decreased between days 3 and 5 post-EPO (Fig 2I). This suggested that the induced antiviral response might be, at least in part, infection-limiting. In contrast, the ΔORF2 RNA signal was already weaker on a per-cell basis compared to WT and ΔORF3 as early as day 3 post-EPO, and it continued to decline thereafter (Fig 2H and 2I). On days 5 and 7 post-EPO, we barely detected any remaining ΔORF2 genomes, as evidenced by a significant decrease in HEV RNA-positive cells compared to day 3 (Fig 2I). This highlights the importance of ORF2 in enabling persistent HEV replication in the presence of an antiviral response.

Interestingly, ΔORF2 RNA was still readily detectable in our bulk RT-qPCR analysis at these time points (Fig 2B). The RNA-FISH assay relies on hybridization of 20 probe pairs to an *ORF2* target region spanning ~1,000 nucleotides [37], whereas RT-qPCR only amplifies a short sequence stretch of 70 nucleotides. Therefore, it is possible that we detected many degraded ΔORF2 genomes by RT-qPCR on days 5 and 7 post-EPO.

Altogether, we found that the presence of ORF2 is critical for enabling persistent HEV replication through a dampened cell-intrinsic antiviral response, likely due to the direct antagonism of TBK1-mediated signaling.

## ORF2 protects HEV replication from antiviral effectors

Next, we aimed to further characterize the increased sensitivity of ΔORF2 replication to the antiviral effectors, as suggested by Fig 2F. To focus solely on the effects of ISGs, we made use of Huh7.5 cells, which only weakly respond to PRR stimulation [38,39]. Accordingly, EPO of Huh7.5 cells with WT and ΔORF2 RNA resulted in comparable viral replication (S7A Fig) due to the lack of a virus-induced antiviral response (S7B Fig). Next, we treated WT- and ΔORF2-electroporated Huh7.5 cells four days post-EPO with either IFNλ1 or IFNα2A for 72 h to induce ISG expression. Comparable ISG induction between WT and ΔORF2 further confirmed that ORF2 does not interfere with JAK/STAT signaling (S7C Fig). Importantly, we detected neither *IFNB1* nor *IFNL1* expression upon IFN treatment of WT- and ΔORF2-electroporated cells. Thus, we excluded the possibility that IFN treatment led to upregulated PRR expression and, consequently, increased ISG induction in a positive feedback loop due to enhanced sensing of HEV RNA. Nonetheless, we observed that ΔORF2 replication was more strongly inhibited by IFN treatment than WT replication (Fig 2J). We concluded that ΔORF2 replication is indeed more sensitive to the antiviral effects of ISGs induced by exogenous IFN treatment, even in the absence of a virus-induced antiviral response.

Altogether, HEV replication becomes more susceptible to the action of ISGs in the absence of ORF2, suggesting additional protective functions of the ORF2 protein, which are independent of its direct interaction with TBK1.

## A balance between HEV replication and the antiviral response at an early infection bottleneck is essential for sustained viral replication

We next sought to identify the time point where the presence of ORF2 is decisive for the fate of HEV replication in authentic infection. To this end, we established a trans-complementation system in hepatoma S10-3 cells ectopically expressing

ORF2 to produce ΔORF2 virus particles ([Fig 3A]). Ectopic ORF2 expression was comparable to the protein level reached by viral replication after EPO of WT HEV RNA into naïve S10-3 cells ([Fig 3B]). We confirmed by WB that ORF3 but not ORF2 was expressed upon EPO of ΔORF2 RNA into naïve cells ([Fig 3B]). We further validated the formation of both

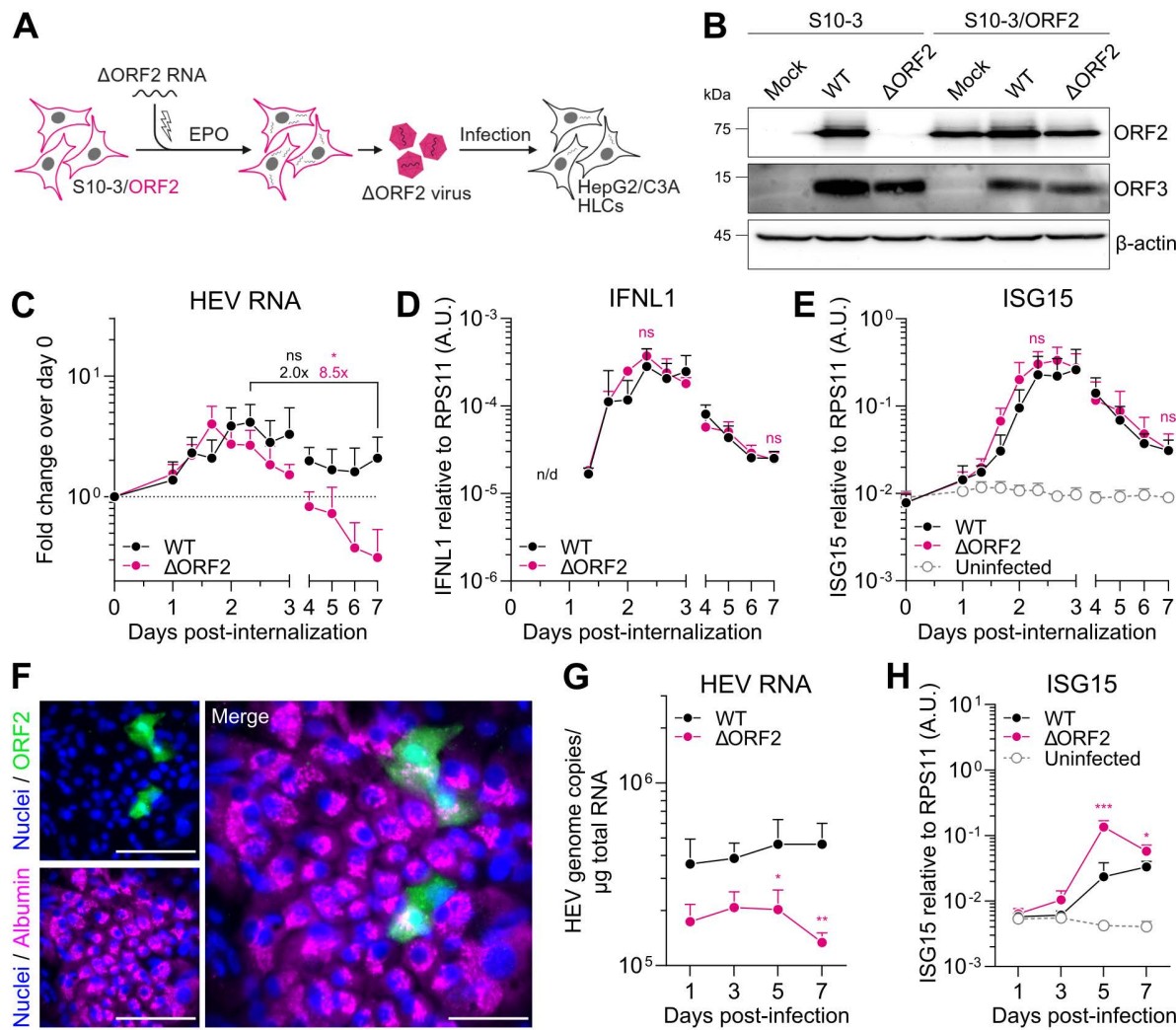

**Fig 3. A balance between HEV replication and the antiviral response at an early infection bottleneck is essential for sustained viral replication.** (A) Schematic of ΔORF2 trans-complementation. S10-3 cells stably expressing HEV ORF2 were electroporated with ΔORF2 HEV RNA to produce virus particles containing the ΔORF2 genome packaged in ORF2 capsid protein expressed by the producer cell. Trans-complemented ΔORF2 virus was used to infect HepG2/C3A cells or stem cell-derived HLCs. Created in BioRender. Dao Thi, V. (2025) https://BioRender.com/u54x394 (B) WB analysis of mock, WT-, or ΔORF2-electroporated S10-3 or S10-3/ORF2 cells for ORF2, ORF3, and β-actin protein expression. (C) Equal genome copies of HEV WT and ΔORF2 virus particles (30 GE/cell) were bound on HepG2/C3A cells for 2 h at 4°C, followed by internalization at 37°C for 8h and inoculum removal (= day 0). RT-qPCR was performed at indicated time points post-internalization to determine fold viral replication over day 0 and (D) *IFNL1* and (E) *ISG15* expression over the housekeeping gene *RPS11* using the $2^{-\Delta Ct}$ method. *IFNL1*, expression was undetectable in the uninfected sample. Statistical analysis of fold changes over time (C) and ΔORF2 compared to WT (D–E) are indicated above the respective time points in the corresponding color. Data show mean±SD of n=3 independent biological experiments. Statistical analysis was performed using unpaired two-tailed Student's t-test of the respective conditions or days independently. *: p<0.05; ns, non-significant. A.U., arbitrary units; n/d, not detectable. (F) Exemplary image of stem cell-derived hepatocyte-like cells (HLCs) infected with HEV WT, fixed and stained for albumin and ORF2 by immunofluorescence. Scale bar, 50 μm. (G) Stem cell-derived HLCs were infected with equal genome copies of HEV WT and ΔORF2 virus particles (1.4 x $10^6$ GE/well) and analyzed by RT-qPCR for HEV RNA genome copies and (H) *ISG15* expression over the housekeeping gene *RPS11* using the $2^{-\Delta Ct}$ method. Data show mean±SD of n=4 biological replicates of two independent HLC differentiations. Statistical analysis of ΔORF2 compared to WT was performed using unpaired two-tailed Student's t-test of each time point independently. *: p<0.05; **: p<0.01; ***: p<0.001.

naked (nHEV, buoyant density of 1.25 g/mL) and quasi-enveloped particles (eHEV, buoyant density of 1.10 g/mL) upon ΔORF2 trans-complementation by performing density gradient ultracentrifugation of cell lysates and supernatants, respectively, and co-detecting HEV RNA and ORF2 protein (S8 Fig). The following experiments were then performed with ΔORF2-nHEV particles, harvested from cell lysates and concentrated by standard ultracentrifugation through a 20% sucrose cushion.

To study the replication dynamics after cell entry in HepG2/C3A cells in a time-resolved manner, we synchronized infection at equal MOI (30 genome equivalents (GE)/cell) of both WT and ORF2-trans-complemented HEV particles. Virus was bound at 4°C for 2 h and internalized at 37°C for 8 h, followed by inoculum removal, which we referred to as day 0 post-internalization. We analyzed viral replication and the antiviral response every 8 h between days 1 and 3 post-internalization and every day (i.e., every 24 h) thereafter by RT-qPCR. Despite infection with the same number of GEs, ΔORF2 HEV RNA levels detected after internalization were consistently ~2.4-fold lower than WT HEV RNA (S9A Fig). This indicated a lower specific infectivity of the trans-complemented ΔORF2 virus particles, potentially due to less efficient progeny assembly. We therefore normalized HEV RNA over day 0 post-internalization (Fig 3C).

Replication of WT and the ΔORF2 mutant peaked at 48 ± 8 h post-internalization, followed by a decline (Fig 3C). Interestingly, while HEV WT replication eventually stabilized between 56 h and day 7 post-internalization (2.0-fold decrease), ΔORF2 replication continued to decrease significantly by 8.5-fold between the two time points. Following the onset of viral replication, we observed an increase in *IFNL1* (Fig 3D) and *ISG15* expression (Fig 3E), which peaked at 56 h and 64 h, respectively, and declined thereafter. *IFNL1* and *ISG15* expression for ΔORF2 were not significantly different from WT at 56 h and on day 7 post-internalization (Fig 3D and 3E). However, the comparatively lower ΔORF2 RNA level after internalization (S9A Fig) and the significant decrease until day 7 compared to WT RNA (Fig 3C) suggested a relatively stronger induction of *IFNL1* and *ISG15* expression in ΔORF2 infection. Accordingly, normalization of the antiviral response to viral replication revealed that ΔORF2 induced a significantly higher *IFNL1* and *ISG15* response per HEV genome (S9B and S9C Fig).

Next, we aimed to validate our findings in a more physiologically relevant hepatocellular system by using human pluripotent stem cell-derived hepatocyte-like cells (HLCs), which are permissive to HEV infection (Fig 3F). We infected HLCs overnight at equal MOI (1.4 x 10$^6$ GE/well) with HEV WT and ΔORF2 virus particles produced by trans-complementation. Similar to our results in infected HepG2/C3A cells, ΔORF2 RNA was lower than WT on day 1 post-infection. Nonetheless, we observed decreased ΔORF2 replication over time (Fig 3G) as well as a significantly stronger induction of *ISG15* compared to HEV WT on days 5 and 7 post-infection (Fig 3H). Similar to our findings made by EPO of HepG2/C3A cells, expression of *ISG15* (Fig 3H), *IFNL1*, and an additional ISG, *IFIT1*, (S9D and S9E Fig) peaked on day 5 post-ΔORF2 infection and decreased again to a similar level as WT on day 7 post-infection.

Overall, our observations suggest that the presence of the ORF2 protein is critical for establishing an equilibrium between viral replication and the antiviral response early in infection, allowing HEV to replicate in different hepatocellular systems despite an antiviral response.

## scRNA-seq analysis reveals globally dampened ISG responses in HEV-infected cells and bystanders in the presence of ORF2

Expression of ISGs can be induced immediately downstream of viral recognition by PRRs, and secreted IFN can lead to amplified ISG induction by auto- and paracrine signaling [11]. By bulk analysis, it remains unclear whether antiviral responses in HEV infection are induced by infected cells, uninfected bystanders, or both. To identify the cellular origin and assess potential differences in the antiviral response in the absence of ORF2, we performed single-cell RNA-sequencing (scRNA-seq) using microfluidics-based 3'-targeted 10x Genomics. To this end, we infected HepG2/C3A cells with HEV WT and trans-complemented ΔORF2 virus particles in a synchronized manner as described above and analyzed the cells at 56 h post-internalization, when both WT and ΔORF2 have reached their replication peak. Since the ΔORF2 virus replicates to low levels at later time points, we only analyzed WT infection on day 7.

First, we performed gene set enrichment analysis (GSEA) of the HEV WT- and ΔORF2-infected samples compared to the uninfected sample at 56 h, using the hallmark gene sets of the Human MSigDB Collections [40]. Interferon alpha and gamma responses were among the most significantly upregulated gene sets, in both HEV WT and ΔORF2 infection, highlighting that the global response was dominated by differential expression of ISGs (Fig 4A and 4B). Enrichment of oxidative phosphorylation and glycolysis indicated increased metabolic activity of the infected samples. A direct comparison of ΔORF2 and WT infection revealed the ISG response, together with oxidative phosphorylation, as the most differently induced gene sets between the two infections (S10A Fig). Analysis of WT infection on day 7 showed that the enriched gene sets were comparable over time and interferon-related responses were still prominently enriched (Fig 4C).

Next, we clustered the uninfected, HEV WT-, and ΔORF2-infected samples at 56 h based on a list of ~400 ISGs, previously published by Schoggins *et al.* [41] (Fig 4D–G). This revealed two major subpopulations in the infected samples (Fig 4E and 4F), also on day 7 (S11A–C Fig). We concluded that the cells located in the cluster overlapping with the uninfected sample were non-responding cells, characterized by unaltered ISG expression. The cells located in the second cluster were characterized by upregulated ISG expression compared to the non-responding cells and were thus classified as responders, from here on called the "active cluster" (Fig 4H). In agreement with the globally enhanced metabolic activity (Fig 4A and 4B), this cluster appeared to be more transcriptionally active, as indicated by increased numbers of expressed genes and transcripts (S10B and S10C Fig). Within the WT- and ΔORF2-infected samples, 28.1% and 17.2% of cells were identified as HEV RNA-positive, respectively (S10D Fig). Interestingly, we observed that most infected cells of both WT- (Fig 4I) and ΔORF2-infected samples (Fig 4J) were located within the active cluster, together with uninfected bystanders. We therefore concluded that both actively infected cells and uninfected bystanders responded by upregulated ISG expression.

Compared with the infected cells in the active cluster, the few infected cells located within the inactive cluster had lower HEV RNA counts (S10E Fig). These cells might be at an earlier stage of infection due to a delayed onset of replication or due to secondary infection, which, however, should be limited at this early time point. On day 7 of WT infection, uninfected bystanders, but also HEV-infected cells, showed enhanced ISG expression and localized to the active cluster (S11D Fig). This provided further evidence that the ISG response is not fully suppressed in actively infected cells by the ORF2-mediated antagonism. Instead, the establishment of an equilibrium between HEV replication and the antiviral response is essential for persistent viral replication in an antiviral environment.

Next, we sought to analyze differences in the transcriptional responses between WT and ΔORF2 infection in more detail. We directly compared the differential expression of ISGs and other genes in a scatter plot and observed ISGs to be among the most prominent and specifically upregulated genes in ΔORF2 infection compared to WT (Fig 4K). This highlighted again that the major differences between WT and ΔORF2 infection affect the antiviral response, but it also suggested that other types of genes may be directly modulated by ORF2. We applied a threshold to highlight genes with potentially higher expression in WT (Fig 4K, black) and ΔORF2 infection (Fig 4K, magenta). Even though we could not find any enriched gene sets among these groups of genes, future studies should assess the impact of the ORF2 protein on their expression in more detail.

We then analyzed the induced ISG signatures, based on the list of ~400 ISGs [42], in more detail and observed a globally stronger ISG response in ΔORF2-infected and -uninfected cells compared to WT infection (Fig 4L). This indicated overall increased IFN production and secretion in ΔORF2 infection. Consulting previous literature on HEV infection of different hepatocellular systems and in chimpanzees [13–15,43], we selected a subset of 30 ISGs for further comparison of the ISG expression pattern in infected cells and bystanders. While the normalized expression level remained increased for the selected ISGs in the absence of ORF2, we observed similar ISG expression patterns across infected cells and uninfected bystanders of both WT and ΔORF2 infection (Fig 4M).

Overall, we identified both actively infected cells and uninfected bystanders as the cellular origins of the ISG response at early and later time points of HEV infection. The absence of ORF2 results in the induction of a globally stronger antiviral

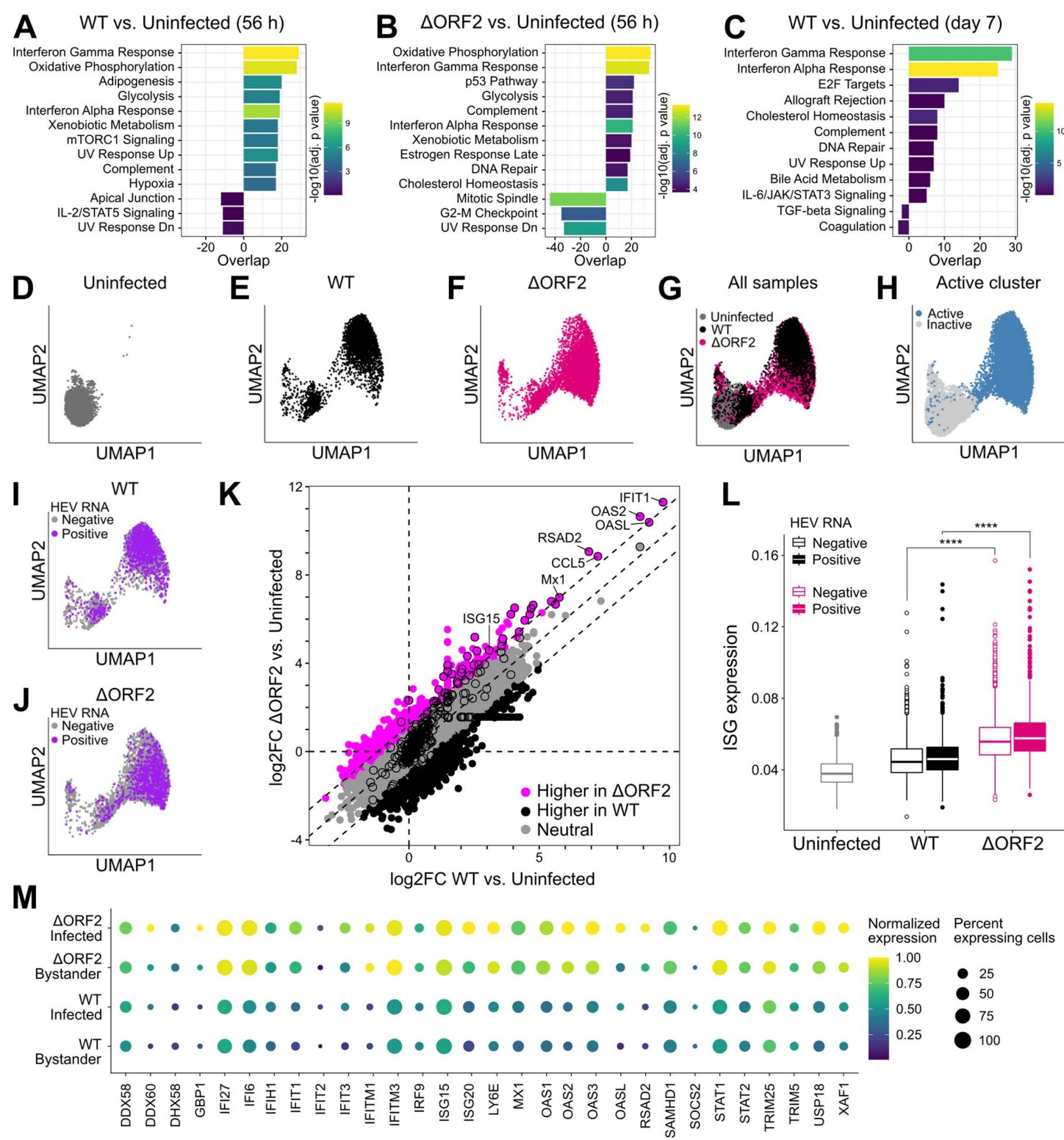

**Fig 4. scRNA-seq analysis reveals globally dampened ISG responses in HEV-infected cells and bystanders in the presence of ORF2.** (A) Uninfected, WT-infected, and ΔORF2-infected HepG2/C3A cells were harvested and processed by microfluidics-based 3'-targeted 10x Genomics at 56 h post-synchronized infection. Uninfected and WT-infected HepG2/C3A cells were also analyzed on day 7 post-synchronized infection. GSEA was

performed using the hallmark gene sets of the Human MSigDB Collections [40], comparing WT-infected with uninfected sample at 56 h post-infection, (B) ΔORF2-infected with uninfected sample at 56 h, and (C) WT-infected with uninfected sample on day 7 post-infection. (D) UMAP projections of uninfected, (E) WT-infected, or (F) ΔORF2-infected HepG2/C3A cells harvested at 56 h post-synchronized infection. UMAPs were generated by clustering of all cells based on a list of ~400 ISGs published previously by Schoggins et al. [41]. (G) Combined UMAP projections of uninfected, WT-, and ΔORF2-infected samples, colored by condition. (H) The cluster of cells with upregulated ISG expression is highlighted in combined UMAP projections of uninfected, WT-, and ΔORF2-infected samples in blue, representing the active cluster. (I) UMAP projections of the WT-infected and (J) ΔORF2-infected samples with indicated binarized HEV RNA counts in purple. (K) Log2FCs of all genes, comparing WT-infected cells vs. cells of the uninfected sample on the x-axis with ΔORF2-infected cells vs. cells of the uninfected sample on the y-axis. Genes were categorized according to their distance to the diagonal: genes with an absolute distance higher than 1.0 were classified as biased towards higher expression in WT (black) or ΔORF2 (magenta). Dots highlighted by a black border are part of the ISG gene list according to Schoggins et al. [42]. (L) Uninfected samples as well as HEV RNA-positive and HEV RNA-negative cells of WT and ΔORF2 infection were assessed for expression of an ISG signature, containing the list of ~400 ISGs according to Schoggins et al. [42]. The area under the curve (AUC) was used to assign scores based on whether the ISGs were expressed within each cell of every sample. Statistical analysis was performed using a two-sided Wilcoxon rank sum test. ****: p < 0.0001. (M) Normalized expression of 30 selected ISGs and the percentage of expressing cells were plotted for WT- and ΔORF2-infected samples, split by actively infected cells (at least one HEV RNA count per cell) and uninfected bystanders.

response, and increased auto- and paracrine IFN signaling lead to enhanced ISG responses in both infected cells and bystanders.

## Discussion

HEV replication has been shown to persist despite a sustained cell-intrinsic antiviral response. However, an in-depth understanding of the underlying mechanisms and the active contribution of a viral antagonism is missing. In this study, we analyzed authentic HEV infection with mutants lacking expression of individual viral proteins at single-cell resolution. We identified a replication-limiting bottleneck imposed by the antiviral response where the presence of ORF2 and, partially, its interaction with the adaptor molecule TBK1, determine whether viral replication declines or equilibrates with the antiviral response (summarized in Fig 5).

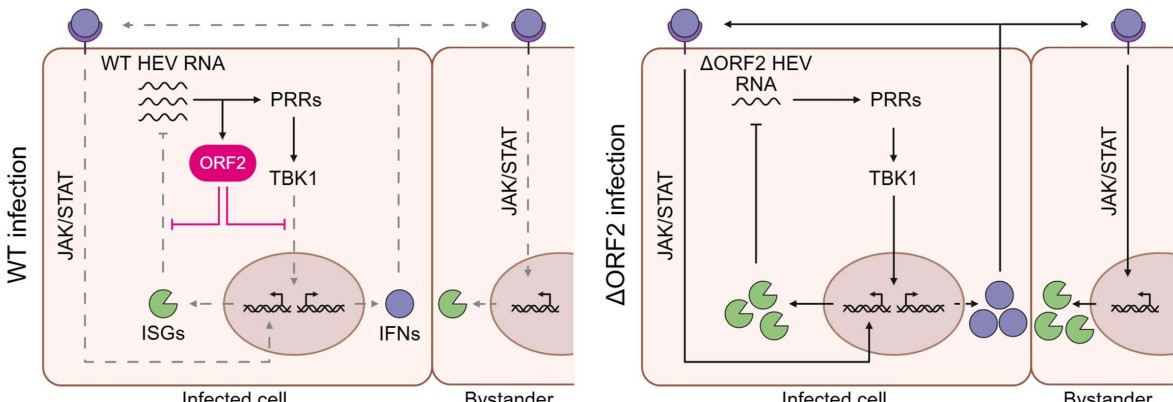

**Fig 5. Working model of HEV ORF2 enabling persistent viral replication by dampening the cell-intrinsic antiviral response.** ORF2 directly interferes with IFN induction downstream of PRR recognition of HEV RNA in WT-infected cells, at least in part through direct interaction with TBK1. Consequently, ISG induction downstream of PRR signaling as well as IFN secretion and thus, autocrine and paracrine ISG induction downstream of the IFN receptor is dampened in infected cells and bystanders, respectively. Additionally, ORF2 shields HEV RNA from the antiviral functions of ISGs. In ΔORF2-infected cells, the lack of ORF2 results in stronger ISG induction downstream of PRR signaling as well as enhanced IFN secretion, leading to a stronger autocrine and paracrine ISG induction downstream of IFN signaling. Viral replication is consequently dampened as a result of the actions of antiviral effectors. Created in BioRender. Dao Thi, V. (2025) https://BioRender.com/ljmqyuk.

## Single-cell analysis uncovers the role of the multifunctional capsid protein for persistent HEV replication amidst an antiviral environment

In the present study, we have shown that the capsid protein ORF2 is a critical viral determinant of HEV replication efficiency. In absence of the ORF2 protein, a stronger cell-intrinsic antiviral response is induced. Importantly, we ruled out that the full-length, single-stranded HEV genome was sensed because the replication-incompetent GNN mutant did not induce IFN or ISG expression (Fig 2C and 2D). Otherwise, sensing of accumulating unpackaged HEV genome due to the missing capsid protein ORF2 could have explained the enhanced antiviral response. A previous study suggested that the 3'-untranslated region (UTR) of single-stranded HEV RNA is sensed by RIG-I [44]. However, as the authors used transfection of short RNA fragments, it remains unclear whether this motif is also sensed in a full-length viral genomic context. We and others have shown that replication is required to induce an antiviral response [14,15], suggesting that the double-stranded replication intermediate might rather serve as a PAMP.

Investigation of the cell-intrinsic antiviral response at the single-cell level revealed that both infected cells and uninfected bystanders are the source of the ISG response in HEV infection, which was globally stronger in both cell types in the absence of ORF2 (Fig 4L and 4M). This per-cell increased antiviral response in ΔORF2 infection only became evident by our scRNA-seq analysis and was otherwise masked in bulk analysis by RT-qPCR. Thus, the ORF2-mediated viral antagonism does not completely suppress antiviral responses within infected cells. Instead, it reduces ISG and IFN induction within infected cells and consequently, further ISG induction via autocrine and paracrine IFN signaling. Overall, ORF2 contributes to dampening of the antiviral response below a critical threshold, allowing viral replication and the antiviral response to reach an equilibrium where they can coexist. We propose that this equilibrium is a prerequisite for persistent HEV infection.

HEV ORF2 appears to be a multifaceted protein with a wide range of functions affecting different host pathways. Aside from its direct involvement in immune evasion downstream of PRR recognition, we speculate that the capsid protein ORF2 exerts additional protective functions. Here, we demonstrated that ORF2 is essential for maintaining the relative resistance of HEV to type I and type III IFN treatment, even in the absence of a virus-induced antiviral response (Fig 2J). Importantly, we were able to exclude a direct inhibition of IFN signaling by ORF2 (Figs 1E and S7C). Therefore, we concluded that the capsid protein additionally contributes to the shielding of replicating RNA from the action of ISGs and, potentially, also from recognition of double-stranded replication intermediates by PRRs. Possibly, ORF2 might protect viral RNA from degradation by ISGs that function as nucleases, either due to the lack of genome packaging or by supporting the formation of the viral replication site, which has not yet been conclusively elucidated [45–48]. Alternatively, ORF2 might antagonize other cellular proteins apart from TBK1, including ISGs that specifically interfere with viral replication.

Moreover, it was proposed previously that the nuclear form of ORF2 may contribute to the dampening of inflammatory cytokine expression [4]. This suggests a direct involvement of ORF2 in host gene modulation, which might further add to the repertoire of the ORF2-mediated evasion of the antiviral response. Our scRNA-seq data suggested subtle differences in the expression of genes other than ISGs between WT and ΔORF2 infection (Fig 4K) and future studies should investigate whether ORF2 might be able to directly or indirectly modulate their expression.

## The direct interaction of HEV ORF2 with TBK1 is critical for antagonizing replication-limiting antiviral responses

There is no convincing evidence for a viral protease encoded by the HEV genome. Indeed, unlike HCV [49] or HAV [50], HEV does not cleave MAVS [14], and in agreement with previously published work [25,26], we found that ORF2 interacts directly with TBK1 to dampen IFN and ISG induction. This could directly impair the establishment of an infection-limiting antiviral state. Here, we identified an early time point at 56 h post-infection where the presence of ORF2 is critical for the progression of authentic HEV infection, facilitating a balance between viral replication and the antiviral response (Fig 3C–E). It should be noted that we employed different hepatocellular systems and different methods for HEV genome or

virus particle delivery in our study. Therefore, the identified 56 h time point is only representative for this specific route of infection. Upon EPO of HepG2/C3A cells and infection of HLCs, we mainly observed differences in the antiviral response between WT and ΔORF2 on day 5. Although the general outcome of ORF2 deletion is the same, the kinetics of replication and antiviral response induction are not directly comparable among these different systems. HEV replication initiates on day 2 [14] in HepG2/C3A cells and on day 3 in HLCs [15] after infection, further complicating a direct comparison of the replication kinetics. Future studies should compare the contribution of pro- and antiviral host factors that may facilitate or impair viral replication in either system.

Lin and colleagues [26] previously suggested that the ARM of ORF2 might be responsible for the inhibition of TBK1-induced phosphorylation of IRF3. Indeed, we could show a reduced interaction between ORF2 and TBK1 upon mutation of the ARM. However, this motif has also been identified as the master regulator of the ORF2 isoforms [4]. Therefore, mutating the ARM could also indirectly affect interference with the antiviral response, as enhanced secretion of glyco-sylated ORF2 along the secretory pathway results in decreased cytosolic and nuclear availability of ORF2 to interact with TBK1 and other unidentified factors. The significance of the different ORF2 isoforms in the interaction with TBK1 and in the protection of viral replication should be investigated in detail. However, as the maturation of these isoforms remains controversial [4,5], the design of conclusive studies may prove challenging.

We found that ORF2 also antagonizes NF-κB-dependent cytokine induction (Figs 1D and S2B and S2C and S2E and S2F) and is hence likely to interfere with additional components other than TBK1. This observation is in agreement with the previously published interaction of ORF2 with beta-transducin repeat containing protein (βTRCP) [27], resulting in reduced proteasomal degradation of the inhibitory complex of NF-κB. Additionally, nuclear translocation of ORF2 has been proposed to affect inflammatory cytokine induction [4]. Considering the numerous suggested ORF2 antagonisms by us and others, we believe that persistent viral replication is likely based on combinatorial interference of ORF2 with different pathways.

Due to the apparent multifunctionality of the ORF2 protein, individual functions are difficult to untangle if specialized domains remain unknown. Indeed, our attempts to predict any putative interaction sites between ORF2 and TBK1 with AlphaFold remained unsuccessful. The only promising prediction was localized to the N-terminus of ORF2. However, this portion of the protein appears to be intrinsically disordered [51]. While AlphaFold has demonstrated success in identifying binding regions between ORF2 and other host factors [52], this particular interaction highlights the challenges of accu-rately predicting interactions involving disordered protein regions [53,54].

## Innate immunity as a potential determinant of HEV chronicity and species tropism

Infections with different HEV genotypes result in distinct clinical outcomes. While HEV-1 infections are acute and self-limiting, HEV-3 infections can become chronic in immunocompromised individuals. The determinants of these intergeno-typic differences remain to be elucidated, but innate and adaptive immune responses are likely critical parameters [19].

We have previously shown that infection with an HEV-1 isolate resulted in a stronger antiviral response than infection with an HEV-3 isolate. Interestingly, HEV-1 but not HEV-3 replication appeared to decrease thereafter [15]. A moderate but prolonged type III IFN response induced by HEV-3 could help to induce tolerance and dampen the effect of the stronger type I IFN response elicited by professional immune cells such as plasmacytoid dendritic cells (pDCs). In agreement, we and others have demonstrated a remarkable resistance of HEV replication to exogenous IFN, once replication is fully established [14,16–18]. Continuous activation of JAK/STAT signaling upon HEV replication has been shown to render the cells refractory to exogenous IFN stimulation [14]. A recent publication further suggested that ORF2 isoforms modulate activation of pDCs upon contact with HEV-infected cells, indicating that this cell type might be contributing to HEV control *in vivo* [34]. Not only do pDCs produce IFN and other cytokines to activate innate immune cells, they also capture and process antigens to initiate adaptive T cell responses. Hence, dampening of the cell-intrinsic antiviral response in infected hepatocytes might affect the crosstalk with professional innate immune cells, which in turn are critical for efficient B and

T cell responses *in vivo*. Naturally, we acknowledge that persistent HEV replication in cell culture is not equivalent to viral persistence *in vivo*. Nonetheless, a thorough understanding of viral perturbation of the antiviral response lays the foundation for more complex studies in the future, including co-culture with immune cells and immunocompetent animal models.

Our results showed no difference in the potential of ORF2 from acute and chronic genotypes to interfere with IRF3- and NF-κB-mediated signaling (Fig 1), which is consistent with previous studies [25,26]. However, HEV-1 is limited to primates and human infections, whereas HEV-3 can infect a wide range of animal species and is mainly transmitted zoonotically to humans. Interaction of ORF2 with TBK1 potentially represents a prime example of the ongoing arms race between virus and host, driving the co-evolution of viral evasion strategies and host defenses. Therefore, future research should investigate the differences in the ability of HEV-1 and HEV-3 ORF2 to counteract TBK1 from different HEV-3 host species, including pig or rabbit. Such studies might contribute to the identification of the determinants of HEV species tropism.

Many open questions remain about the mechanisms of HEV persistence in cell culture and *in vivo*. In this study, we have described at least two different ORF2-mediated immune evasion strategies, affecting TBK1-dependent IFN and ISG induction and protection of viral RNA from antiviral effectors. ORF2-mediated immune evasion becomes essential at a replication-limiting bottleneck early in infection, mediated by the antiviral response, which is decisive for establishment of persistent viral replication. We have also laid the groundwork for future studies to focus on the role of intergenotypic differences in the antiviral response and on the crosstalk with immune cells. This could eventually lead to a better understanding of the determinants of acute and chronic manifestations of HEV infection, which will be valuable in identifying novel therapeutic regimens.

## Materials and methods

### Ethics statement

This project (AZ: 3.04.02/0137) has been approved by the Zentrale Ethik-Kommission für Stammzellenforschung of the Robert Koch Institute (RKI) and fulfils all legal requirements according to the German Stem Cell Act.

### Standard cell culture

The human hepatoma cell lines HepG2/C3A (ATCC HB-8065), Huh7.5 (a kind gift from Charles Rice, The Rockefeller University, New York City, USA), S10-3 (a kind gift from Suzanne Emerson, NIH, Bethesda, USA), and derived S10-3/ORF2 as well as the human embryonic kidney cell line HEK293T (ATCC CRL-3216) were cultured in Dulbecco's Modified Eagle Medium (DMEM, Gibco, high glucose, GlutaMAX supplement) with 10% fetal bovine serum (FBS, Capricorn) and 1% penicillin/streptomycin (pen/strep), referred to as complete DMEM (cDMEM). HepG2/C3A cells were grown on collagen-coated cell culture vessels. A549-derived cell lines were cultured in DMEM (Gibco, high glucose) with 10% FBS (Pan Biotech), 1% pen/strep, and 1% non-essential amino acids (NEAA). Cell lines with ectopic protein expression were produced by standard lentiviral transduction, selected, and continuously cultured under respective antibiotic selection pressure. Cells were maintained at 37°C in 95% humidity and 5% $CO_2$ atmosphere. All cell lines used in this study routinely tested negative for mycoplasma.

### Stimulation of PRR-overexpressing A549-derived cell lines

$1 \times 10^5$ A549-derived cells were seeded in 24-well plates. The next day, cells were infected with Mengo-Zn virus [55] (MOI 1) for 24 h, Sendai virus (SeV, MOI 0.75; prepared from allantoic fluid of embryonated chicken eggs, a kind gift from Rainer Zawatzky, German Cancer Research Center (DKFZ), Heidelberg, Germany) for 4 h, or supernatant-fed with high molecular weight (HMW) poly(I:C) (InvivoGen) at 50 µg/mL for 24 h in DMEM containing 2% FBS, 1% NEAA, and 1% pen/strep. Stimulation with TNF (Abcam) at 10 ng/mL or IFNβ (R&D Systems) at 200 IU/mL was performed in cDMEM plus 1% NEAA for 8 h. At respective time points, cells were lysed for RNA extraction with Monarch Total RNA Miniprep Kit (New England Biolabs).

### Generation of HEV ΔORF2, ΔORF3, and GNN mutants

Mutants were generated in the plasmid pBlueScript SK(+) encoding the HEV-3 Kernow-C1/p6 sequence (a kind gift from Suzanne Emerson, NIH, Bethesda, USA; pBSK-HEV-p6, GenBank accession number: JQ679013.1) or the pUC-HEV-83-2-27 plasmid [36] (a kind gift from Koji Ishii and Takaji Wakita, National Institute of Infectious Diseases, Tokyo, Japan; GenBank accession number: AB740232) by site-directed mutagenesis. Mutations were introduced by overlap extension PCR with Phusion or Q5 polymerase (New England Biolabs) using the primers listed in S1 Table, followed by restriction enzyme digestion (New England Biolabs), ligation with T4 DNA Ligase (New England Biolabs), transformation of JM109 competent cells (Promega), and validation by Sanger DNA sequencing.

### *In vitro* transcription and EPO of HEV WT, ΔORF2, ΔORF3, and GNN mutants

pBSK-HEV-p6-derived and pUC-HEV-83-2-27-derived plasmids were linearized with MluI or HindIII, respectively, and RNA was *in vitro* transcribed using the mMESSAGE mMACHINE T7 kit (Invitrogen). 4 x $10^6$ HepG2/C3A cells were electroporated at 270 V and 975 µF with 10 µg of IVT HEV RNA in cytomix (120 mM KCl, 0.15 mM CaCl$_2$, 10 mM KPO$_4$, 25 mM HEPES, 2 mM EGTA, and 5 mM MgCl$_2$), supplemented with adenosine triphosphate (ATP) and glutathione (GT). After resuspension in cDMEM, HepG2/C3A cells electroporated with the Kernow-C1/p6 strain or derived mutants were mixed 1:1 with mock-electroporated cells. 2 x $10^5$ cells/well were seeded on 24-well plates for analysis by RT-qPCR and 1 x $10^5$ cells/well on 48-well plates for RNA-FISH. One day post-EPO, all wells were washed twice with PBS to remove inoculum and cDMEM was replaced. Cell lysates were harvested for RNA extraction using the Universal RNA kit (Roboklon) according to manufacturer's instructions. On days 1, 3, and 5 post-EPO, cDMEM was replaced and 6 µM TBK1 inhibitor BX795 (InvivoGen) or corresponding DMSO vehicle control (0.06%) was added 48 h prior to harvesting. Samples for RNA-FISH were fixed at respective time points with 4% paraformaldehyde (PFA, Electron Microscopy Sciences). For analysis of STAT1 and pSTAT1 levels, lysates were harvested on days 1, 3, 5, and 7 post-EPO, and 15 µg of protein, determined by BCA assay (Thermo Fisher Scientific), were subjected to SDS-PAGE and WB analysis. IFN secretion was measured in supernatants collected on respective days using a multiplex immunoassay (U-PLEX Interferon Combo (human), Meso Scale Discovery).

### EPO of Huh7.5 cells and IFN treatment

Huh7.5 cells were electroporated with HEV WT or ΔORF2 IVT RNA as described above. For time-course analysis, electroporated Huh7.5 cells were resuspended in cDMEM, seeded in 24-well plates, and harvested for RT-qPCR analysis at respective time points. For IFN treatment, cells were plated in a T75 cell culture flask after EPO. On day 3 post-EPO, cells were reseeded to 24-well plates at ~80% confluency. One day later, cells were treated with 10,000 IU/mL IFNα2A (Tebubio) or 10 ng/mL IFNλ1 (Gibco). IFNs were replenished daily until cells were lysed for RNA extraction with the Universal RNA kit (Roboklon) and RT-qPCR on day 7 post-EPO.

### ΔORF2 virus production by trans-complementation

S10-3/ORF2 cells were electroporated with IVT ΔORF2 RNA and expanded on day 3 post-EPO. Intracellular nHEV particles were harvested 7 days post-EPO from cell lysates by four repeated freeze-thaw cycles and pelleted by ultracentrifugation at 28,000 rpm for 3 h through a 20% sucrose cushion. HEV WT and ΔORF2 genome copies were determined by RT-qPCR following TRIzol extraction from the resuspended virus prep.

### Synchronized, time-resolved HEV infection

6 x $10^4$ HepG2/C3A cells were seeded on collagen-coated 24-well plates. The next day, cells were infected with equal GEs of HEV WT or ΔORF2 virus (30 GE/cell). Virus was bound at 4°C for 2 h in Minimum Essential Medium (MEM,

Gibco), supplemented with 10% FBS and 1% pen/strep, followed by internalization at 37°C for 8 h. After internalization, inoculum was removed, cells were washed twice with PBS, and cDMEM was added. Samples for RT-qPCR were lysed at respective time points with TRIzol (Invitrogen) for RNA extraction.

## Generation of human pluripotent stem cell-derived HLCs and HEV infection

The hESC cell line RUES2 [56] was cultured in mTeSR1 (STEMCELL Technologies) on cell culture plates coated with Matrigel (Corning). RUES2 cells were differentiated to definitive endoderm (DE) using the STEMdiff Definitive Endoderm Differentiation Kit (STEMCELL Technologies) according to manufacturer's instructions. As described previously [15], DE cells were reseeded and differentiated to hepatocyte progenitors for five days in basal Rosewell Park Memorial Institute (RPMI) 1640 medium with HEPES (Gibco), supplemented with B-27 custom supplement (Gibco), GlutaMAX (Gibco), NEAA (Gibco), and pen/strep (Gibco), additionally containing the human growth factors bone morphogenetic protein 4 (BMP4, PeproTech) and fibroblast growth factor basic (FGFb, Gibco). Immature hepatocytes were obtained after five days in basal RPMI containing human hepatocyte growth factor (HGF, PeproTech). Maturation into hepatocytes was achieved by final differentiation in the supplemented Hepatocyte Culture Medium BulletKit (HCM, Lonza; no HEGF component) containing human oncostatin M (OSM, R&D Systems). Mature HLCs were infected with equal genome copies of HEV WT and ΔORF2 ($1.4 \times 10^6$ GE/well) in HCM/OSM medium. The next day, the inoculum was removed and cells were washed twice with Dulbecco's Balanced Salt Solution (DPBS, Gibco). HCM/OSM medium was replenished every two days. Cells were fixed with 4% PFA or lysed for RNA extraction with TRIzol (Invitrogen) at respective time points.

## 3'-targeted 10x Genomics and Illumina sequencing

$8 \times 10^4$ HepG2/C3A cells were seeded on collagen-coated 24-well plates, and synchronized HEV infection was performed as described above. At 56 h and day 7 post-internalization, cells were trypsinized for 10 min, gently resuspended, and singularized by pipetting thrice through a 70 μm cell strainer. Two wells of each sample were combined. Single-cell suspensions were washed once and resuspended in 0.04% bovine serum albumin (BSA, Carl Roth) in PBS. Cell suspension was counted and inspected for cell death using trypan blue. Single-cell suspensions were loaded onto the 10x Chromium controller according to manufacturer's instructions of the Chromium Next GEM Single Cell 3′ Kit v3.1 (10x Genomics) with a targeted cell recovery of 5,000. Sequencing libraries were prepared according to the manufacturer's instructions. Briefly, GEMs were generated, reverse transcription was performed, GEMs were cleaned up, and cDNA was amplified and cleaned up with SPRIselect beads (Beckman Coulter). Quantification and quality control were performed using Qubit (Thermo Fisher Scientific) and the 5200 Fragment Analyzer System (Agilent Technologies). Fragmentation, end repair, and A-tailing were followed by SPRIselect-based cleanup, adaptor ligation, sample indexing, and again, SPRIselect cleanup. Quantification and quality control were performed with the 5200 Fragment Analyzer System (Agilent Technologies) and the NEBNext Library Quant Kit for Illumina (New England Biolabs). Resulting libraries were pooled and sequenced on Illumina NextSeq550 (high-output mode, paired-end, 150 cycles).

## scRNA-seq data analysis

Fastq files were aligned and counted using the Cell Ranger 7.1.0 pipeline. GRCh38-3.0.0 was used as a human genome reference. For the GSEA, we first performed a differential expression analysis (DEA) using Wilcoxon rank sum test as implemented in the FindMarkers function from Seurat, filtering out genes expressed in less than 5% of cells in the compared conditions and with p-adjusted values < 0.05. Then, the GSEA was performed with the enrichR v3.2 R package. As a reference, we used the hallmark gene sets of the Human MSigDB Collections [40]. For clustering and UMAP projections, we used only a subset of genes based on a list of ISGs reported by Schoggins *et al.* [41]. Counts were scaled and log1-normalized. The UMAP projections were calculated over the first 20 dimensions of the PCA, setting the following

parameters: min.dist = 0.5 and n.neighbors = 200. In order to identify clusters associated with transcriptionally active/inactive states, we set a resolution of 0.1 in the FindClusters function and then we inspected the expression of ISGs in the identified clusters. Binarized ORF2 expression was carried out by defining positive cells as those expressing at least one HEV RNA count. Comparison of HEV RNA counts across clusters was done using log1 normalized viral ORF2 read counts. To separately compare infected cells in WT and ΔORF2 vs. uninfected sample, we used the output log2FCs from FindMarkers, showing all genes without any filtering to find overall trends. Genes were categorized based on the distance to the diagonal and delta defined as the difference in log2FCs between WT and ΔORF2. We used a threshold of delta = 1.0 to define genes with higher expression in WT or ΔORF2. ISG expression across populations was performed using AUC values to score cells based on the Schoggins signature mentioned above, implemented with the AUCell 1.24.0 R package. Statistical analysis for this data set was performed using two-sided Wilcoxon rank sum tests. To generate dot plots, we used the Dotplot built-in function from Seurat, and values were z-normalized by gene. R 4.3.3 and Seurat 5.0 were used for the entire analysis. The data is accessible under the GEO accession number GSE288400.

## Quantitative (real-time) reverse transcription PCR (RT-qPCR)

RNA was extracted from cell lysates either with the Universal RNA kit (Roboklon), Monarch Total RNA Miniprep Kit (New England Biolabs), or with TRIzol reagent (Invitrogen), following respective manufacturer's instructions. cDNA was synthesized using the iScript cDNA Synthesis Kit (Bio-Rad) or the High Capacity cDNA Reverse Transcription Kit (Thermo Fisher Scientific) and diluted. qPCR was then performed with iTaq Universal SYBR Green Supermix (Bio-Rad) on a CFX96 Real-Time PCR Detection System (Bio-Rad) using the primers listed in S1 Table. Absolute HEV genome copies were calculated from an HEV standard curve, produced by serial 10-fold dilutions of a 10 ng/µL-concentrated pBSK-HEV-p6 plasmid. *IFNL1*, *ISG15*, and *IFIT1* expression in HepG2/C3A cells, HLCs, and Huh7.5 cells were normalized over the housekeeping gene *RPS11* using the $2^{-\Delta Ct}$ method or additionally normalized to HEV RNA using the $2^{-\Delta\Delta Ct}$ method. For *IFNB1*, *TNFAIP3*, *IFIT1*, and *IL6* expression in A549-derived cells, relative expression over the housekeeping gene *GAPDH* was calculated using the $2^{-\Delta Ct}$ method.

## Co-immunoprecipitation

$1.2 \times 10^7$ HEK293T cells were seeded on 10-cm cell culture dishes coated with 100 µg/mL poly-L-lysine (Sigma-Aldrich). The next day, cells were co-transfected with 10 µg of plasmid encoding ORF2-HA or HA-tagged ORF2 mutants and 10 µg of plasmid encoding TBK1-V5 in cDMEM without pen/strep. Polyethyleneimine (PEI, Polysciences) was used for transfection at a DNA:PEI ratio of 1:3. Medium was changed after 5–6 h. After 24 h, cells were washed twice with cold PBS and harvested by scraping in cold PBS containing 0.5x cOmplete Mini Protease Inhibitor Cocktail (Roche). Cells were pelleted and resuspended in 1 mL cold lysis buffer (25 mM Tris-HCl, pH 7.5, 150 mM NaCl, 1 mM EDTA, 1% NP-40, 5% glycerol) containing 1x cOmplete Mini Protease Inhibitor Cocktail (Roche) and incubated on ice for 30 minutes. After removal of DNA, 6% of the lysate was set aside as input and boiled with Laemmli sodium dodecyl sulfate (SDS) sample buffer at 95°C for 10 min. 20 µL Pierce Anti-HA Magnetic Beads (Thermo Fisher Scientific) were added to each sample and incubated under rotation for 2 h at 4°C. Beads were washed thrice with lysis buffer containing 1x cOmplete Mini Protease Inhibitor Cocktail (Roche) and eluted in Laemmli SDS sample buffer by boiling at 95°C for 10 min.

## Western blot

Co-IP samples were prepared as described above. Electroporated S10-3 and S10-3/ORF2 cells and A549-derived cell lines were lysed in Pierce RIPA buffer (Thermo Fisher Scientific) with 1x cOmplete Mini Protease Inhibitor Cocktail (Roche) on ice for 30 min, supplemented with Laemmli SDS sample buffer, and boiled at 95°C for 10 min. Lysis buffer for phospho-blot samples additionally contained 1x Halt Phosphatase Inhibitor (Thermo Fisher Scientific). For detection of ORF3, samples were immediately run on SDS-PAGE gels. Proteins were transferred and membranes were blocked with

5% milk (Carl Roth) in PBS/0.1% Tween-20 (PBS-T) or 5% BSA (Carl Roth) in TBS-T for phospho-blots. The following antibodies were incubated overnight in 5% milk/PBS-T or 5% BSA/TBS-T at 4°C: mouse-anti-β-actin, 1:4000 (Sigma-Aldrich, cat. no. A2228); mouse-anti-ORF2 1E6, 1:500 (Merck, cat. no. MAB8002); mouse-anti-ORF3, 1:25 (University of Geneva Antibody Facility, cat. no. ABCD_RB198); rabbit-anti-HA, 1:1000 (Cell Signaling Technology, cat. no. 3724); rabbit-anti-V5, 1:1000 (Cell Signaling Technology, cat. no. 13202); rabbit-anti-STAT1, 1:500 (Cell Signaling Technology, cat. no. 9172); rabbit-anti-pSTAT1, 1:500 (Cell Signaling Technology, cat. no. 9167). After three washes with PBS-T or TBS-T, membranes were incubated with horseradish peroxidase (HRP)-coupled anti-mouse or anti-rabbit secondary antibodies (Jackson Immunoresearch, 1:4000) for 1 h at room temperature. Membranes were washed thrice with PBS-T or TBS-T and once with PBS or TBS, and chemiluminescent signal was developed using Pierce ECL Western Blotting Substrate (Thermo Fisher Scientific).

### RNA fluorescence *in situ* hybridization and quantification

Fixed, electroporated HepG2/C3A cells were stained using the RNAscope Multiplex Fluorescent V2 Assay (ACDBio), following the manufacturer's protocol, starting from the incubation step with hydrogen peroxide. HEV RNA was specifically targeted with the V-HEV-p6-ORF2 probe (ACDBio, cat. no. 586651), detecting both genomic and subgenomic HEV RNA. Cells were counterstained with Hoechst (Thermo Fisher Scientific), and five randomly selected fields of view per condition and biological experiment were imaged using an inverted Nikon Eclipse Ts2-FL widefield fluorescence microscope. These images were quantified by segmentation of the HEV RNA signal in ilastik [57], followed by overlay with the nuclear signal, segmented in CellProfiler [58]. The percentage of HEV RNA-positive cells was obtained using CellProfiler. For representation, images were merged and adjusted with Fiji [59].

### Immunofluorescence staining

Samples were fixed with 4% PFA and permeabilized/blocked in 10% goat serum (MP Biomedicals), 1% bovine serum albumin (BSA, Carl Roth), and 0.1% Triton X-100 in PBS. Primary antibodies were incubated overnight at 4°C in blocking/permeabilization solution with the following dilutions: rabbit-anti-ORF2, 1:6000 (a kind gift from Rainer Ulrich, Friedrich-Loeffler-Institut, Riems, Germany [60]); mouse-anti-albumin, 1:1000 (Tebubio, cat. no. CL2513A). Samples were washed thrice with PBS and incubated with Alexa Fluor-conjugated secondary antibodies (1:1000, Thermo Fisher Scientific) for 1 h at room temperature. After washing thrice, cells were counterstained with Hoechst (1:1000, Thermo Fisher Scientific) and imaged using an inverted Nikon Eclipse Ts2-FL widefield fluorescence microscope. Images were analyzed and merged with Fiji [59].

### IFNβ ELISA

Supernatants from A549 cells challenged with Mengo-Zn virus, Sendai virus (SeV), and poly(I:C) supernatant feeding for 24 h were collected and IFNβ secretion was measured using a bioluminescent human IFNβ ELISA (LumiKine™ Xpress hIFN-β 2.0, Invivogen) according to manufacturer's instructions.

### Electroporation of A549 cells

$2 \times 10^6$ cells were pelleted, resuspended in 200 μL cytomix (120 mM KCl, 0.15 mM $CaCl_2$, 10 mM $KPO_4$, 25 mM HEPES, 2 mM EGTA, and 2 mM $MgCl_2$), and transferred to a 0.2 cm cuvette containing 500 ng high molecular weight (HMW) poly(I:C) (Invivogen) or control poly(C) (Sigma-Aldrich). EPO was performed at 166 V and 500 μF using the Gene Pulser Xcell modular electro-transfection system (Bio-Rad). Transfected cell suspensions were transferred to pre-warmed DMEM, centrifuged, and resuspended in 4 mL cDMEM plus 1% NEAA. 500 μL of the final cell suspensions were seeded in 24-well plates.

## Cell viability (MTS) assay

After EPO of HepG2/C3A with HEV WT RNA or respective mutants, $3 \times 10^4$ cells were seeded in triplicates on 96-well plates. On days 1, 3, and 5 post-EPO, cDMEM was replaced and 6 μM TBK1 inhibitor BX795 (InvivoGen) or corresponding DMSO vehicle control were added to the wells dedicated to analysis 48 h later. At the respective time points, CellTiter 96 AQueous One Solution Cell Proliferation Assay (Promega) was performed according to manufacturer's instructions. Cells were incubated for 1 h and absorbance at 490 nm was measured using a plate reader (Tecan).

## Computational prediction of ORF2 and TBK1 interaction using AlphaFold 2.3

To predict interactions between ORF2 and TKB1, we constructed 13 distinct systems, varying the chunking strategy and oligomerization states of both proteins. For each system, five structural models were generated using three different random seeds, resulting in a total of 195 models. These models were produced using an in-house implementation of ColabFold [61], employing AlphaFold-Multimer 2.3 weights [62,63] and a multiple sequence alignment computed with the MMSEQS2 webserver [64–67]. Model quality was assessed using pLDDT [68], Predicted Aligned Error, and actifPTM (ACTual InterFace PTM) scores [69]. Additionally, we computed pdock, pdock2, and LIS scores for each model using the AF_analysis package (https://github.com/samuelmurail/af_analysis). Contact maps were generated in-house using the MDTraj Python library [70]. A contact between two amino acids was defined as the presence of at least two atoms within a distance of 5 Å. All scripts used and models produced in this study are available on Zenodo (https://doi.org/10.5281/zenodo.14751497).

## Analysis of nHEV and eHEV particles by density gradient ultracentrifugation

nHEV particles were harvested on day 7 post-EPO from cell lysates of S10-3/ORF2 cells electroporated with ΔORF2 RNA by repeated freeze-thaw cycles and immediately subjected to density gradient ultracentrifugation using Opti-Prep (Sigma-Aldrich) as described previously [37]. For eHEV, cell culture supernatants were concentrated by ultra-centrifugation at 28,000 rpm for 2 h at 4°C. The pellet was resuspended in PBS and subjected to density gradient ultracentrifugation using OptiPrep as described previously [37]. Twelve fractions each were collected, the density was measured, and RNA was extracted using TRIzol (Invitrogen) to determine HEV genome copies by RT-qPCR. The remaining fractions were concentrated through 100 kDa centrifugal filter units (Millipore) and subjected to WB analysis.

## Statistical analysis

Graphs were generated and statistical analysis was performed using GraphPad Prism 8 or R 4.3.3 for the scRNA-seq analysis. Statistical tests and p-values are listed in the respective figure legends for each panel.

## Supporting information

**S1 Fig. Validation of protein expression in A549-derived cell lines and assessment of IFNβ protein secretion upon stimulation.** (A) A549 cells harboring knockouts of the PRRs RIG-I and MDA5 and ectopically expressing a single PRR (MDA5, RIG-I, or TLR3) together with either HEV-3 ORF2, HEV-1 ORF2, ORF3, or GFP were analyzed for ORF2 or (B) ORF3 protein expression by Western blot, together with the loading control β-actin. (C) A549-derived cell lines were challenged with either Mengo-Zn virus at MOI 1, (D) Sendai virus (SeV) at MOI 0.75, or (E) 50 μg/mL poly(I:C) supernatant feeding for 24 h. Supernatant was collected and IFNβ was measured by enzyme-linked immunosorbent assay (ELISA). Numbers indicate fold reductions compared to GFP. Data shown mean ± SD of n = 3 independent biological experiments. Statistical analysis was performed using two-way ANOVA. *, $p < 0.05$; ****: $p < 0.0001$; ns, non-significant. (TIFF)

**S2 Fig. ORF2 dampens the strength, but not the kinetics, of IRF3- and NF-κB-dependent antiviral and inflammatory signaling.** (A) A549 cells harboring knockouts of the PRRs RIG-I and MDA5 and ectopically expressing MDA5 and either GFP or HEV-3 ORF2 were electroporated with poly(I:C) and analyzed for *IFNB1*, (B) *TNFAIP3*, and (C) *IL6* expression at indicated time points, relative to the housekeeping gene *GAPDH* using the $2^{-\Delta Ct}$ method. (D) As a control, A549-derived cells were electroporated with poly(C) and analyzed for *IFNB1*, (E) *TNFAIP3*, and (F) *IL6* expression at indicated time points, relative to the housekeeping gene *GAPDH* using the $2^{-\Delta Ct}$ method. Data shown mean ± SD of n = 3 independent biological experiments.
(TIFF)

**S3 Fig. AlphaFold 2.3 modeling of the ORF2-TBK1 interaction.** (A) actifPTM scores assessing predicted interactions between segments of ORF2 and TBK1. Higher scores indicate stronger predicted interactions. (B) HEK293T cells were transfected with ORF2-HA, ORF2-WRD/AAA-HA, or ORF2-2R/2A-HA and V5-tagged TBK1 and lysed 24 h post-transfection. Anti-HA co-IP and WB analysis for TBK1 (anti-V5 staining), ORF2 (anti-HA staining), and β-actin were performed. Representative blot of n = 2 independent biological experiments. (C–D) Structural analysis of the top five ranked models for the interaction between ORF2 residues 1–128 (red) and a TBK1 dimer. Panel C highlights the molecular environment surrounding ORF2 W87, color-coded by domain (TBK1 kinase domain in turquoise, C-terminal domain in blue). Panel D depicts the same region colored according to pLDDT scores, reflecting model confidence (orange – very low; yellow – low; cyan – high; blue – very high). Modeling convergence of the models and the chemical environment of W87 and R88 suggest potential for stabilizing cation-π interactions. (E) Predicted Aligned Error (PAE) for the highest-ranking ORF2 1–128 vs. TBK1 dimer model. Low PAE values in the N-terminal region of ORF2 1–128, particularly with respect to TBK1 chains B and C, indicate a likely accurate placement of this region relative to the TBK1 monomers.
(TIF)

**S4 Fig. Normalized *IFNL1* and *ISG15* expression over HEV RNA and effect of the TBK1 inhibitor BX795 on cell viability and *ISG15* expression.** (A–B) Normalized data of Fig 2C and 2D over HEV RNA: Expression of (A) *IFNL1* and (B) *ISG15* upon EPO of HepG2/C3A cells with HEV WT, ΔORF2, or ΔORF3 were determined relative to the housekeeping gene *RPS11* and normalized to HEV RNA using the $2^{-\Delta\Delta Ct}$ method. Data show mean ± SD of n = 3 independent biological experiments. Statistical analysis was performed using one-way ANOVA of each time point independently, and comparisons to WT are indicated above the respective time points in the corresponding colors. *: $p < 0.05$; **: $p < 0.01$; ns, non-significant. A.U., arbitrary units; norm., normalized. (C) HepG2/C3A cells were either mock-electroporated or electroporated with HEV WT, ΔORF2, or ΔORF3 RNA and additionally treated with 6 µM of the TBK1 inhibitor BX795 (TBKi) or respective DMSO vehicle control 48 h prior to the time point of harvest. Cell viability was determined on days 3, (D) 5, and (E) 7 post-EPO using the CellTiter 96 AQueous One Solution Cell Proliferation Assay (Promega). Data was normalized to the respective DMSO controls. Data show mean ± SD of n = 3 independent biological experiments. (F) Cell lysates from Fig 2E–G were analyzed for *ISG15* expression relative to *RPS11* using the $2^{-\Delta Ct}$ method. The dashed line indicates the mean of *ISG15* basal expression in mock-electroporated cells under TBKi treatment across days 3, 5, and 7. Data show mean ± SD of n = 3 independent biological experiments. *: $p < 0.05$; **: $p < 0.01$; ***: $p < 0.001$; ns, non-significant.
(TIFF)

**S5 Fig. HEV ΔORF2 induces stronger IFNλ1 secretion and STAT1 phosphorylation than WT in HepG2/C3A cells.** (A) HepG2/C3A cells were electroporated with HEV WT or ΔORF2 and mixed 1:1 with mock-electroporated cells. Supernatants were analyzed for secreted IFNλ1 protein on indicated days post-EPO using a multiplex immunoassay (U-PLEX Interferon Combo (human), Meso Scale Discovery). LOD indicates the lower limit of detection. Data show mean ± SD of n = 3 independent biological experiments. Statistical analysis of ΔORF2 over WT was performed using unpaired two-tailed

Student's t-test of each time point independently. *: $p < 0.05$; **: $p < 0.01$; ns, non-significant. (B) HepG2/C3A cells electroporated with HEV WT or ΔORF2, mixed 1:1 with mock-electroporated cells, were harvested on indicated time points post-EPO for WB analysis. Samples were analyzed for STAT1, pSTAT1, ORF2, and β-actin protein expression. Representative blot of n = 3 independent biological experiments.
(TIFF)

**S6 Fig. The HEV-3 strain 83-2-27 replicates to lower levels and induces a stronger antiviral response in HepG2/C3A cells.** (A) HepG2/C3A cells were electroporated with RNA of the HEV-3 strain 83-2-27 and the derived ΔORF2 mutant. Samples were harvested at indicated time points post-EPO and analyzed by RT-qPCR for HEV RNA genome copies and (B) *ISG15* expression relative to the housekeeping gene *RPS11* using the $2^{-\Delta Ct}$ method. Data show mean ± SD of n = 3 independent biological experiments. Statistical analysis of ΔORF2 over WT was performed using unpaired two-tailed Student's t-test of each time point independently. *: $p < 0.05$; **: $p < 0.01$; ns, non-significant.
(TIFF)

**S7 Fig. Assessment of HEV WT and ΔORF2 replication and ISG expression upon IFN treatment in Huh7.5 cells.** (A) Huh7.5 cells were electroporated with HEV WT, ΔORF2, or replication-incompetent GNN RNA and analyzed at indicated time points post-EPO for HEV RNA by RT-qPCR. Data show mean ± SEM of n = 2 independent biological experiments. (B) Electroporated Huh7.5 cells from (A) were analyzed by RT-qPCR for *IFIT1* expression relative to the housekeeping gene *RPS11* using the $2^{-\Delta Ct}$ method. Data show a single biological experiment. (C) Cell lysates from Fig 2J, treated with IFNλ1 or IFNα2A, were analyzed for *ISG15* expression on day 7 post-EPO, relative to the housekeeping gene *RPS11* using the $2^{-\Delta Ct}$ method. Data show mean ± SD of n = 4 biological repeats from two independent biological experiments. Statistical analysis was performed using two-way ANOVA. ns, non-significant.
(TIFF)

**S8 Fig. ΔORF2 trans-complementation gives rise to both nHEV and eHEV particles.** (A) Cell lysates containing nHEV and (B) cell culture supernatants containing eHEV of S10-3/ORF2 cells electroporated with ΔORF2 RNA were purified through density gradient ultracentrifugation. Gradients were harvested in twelve fractions. The buoyant density of each fraction was measured, and HEV GE/mL were determined in each fraction (upper panels). The fractions were further concentrated and analyzed for the presence of ORF2 protein by WB (lower panels). Data show the results of a single ΔORF2 virus production.
(TIFF)

**S9 Fig. HEV RNA and antiviral response gene expression upon WT and ΔORF2 infection in HepG2/C3A cells and HLCs.** (A) Non-normalized data of Fig 3C: Equal genome copies of HEV WT and ΔORF2 virus particles (30 GE/cell) were bound on HepG2/C3A cells for 2 h at 4°C prior to internalization at 37°C for 8 h, followed by removal of inoculum (= day 0). RT-qPCR was performed at indicated time points post-internalization to determine HEV genome copies. (B) Normalized data of Fig 3D and 3E over HEV RNA: *IFNL1* and (C) *ISG15* expression relative to the housekeeping gene *RPS11* were additionally normalized over HEV RNA using the $2^{-\Delta\Delta Ct}$ method. Statistical analysis of fold changes of ΔORF2 over WT are indicated above the respective time points. Data show mean ± SD of n = 3 independent biological experiments. Statistical analysis was performed using unpaired two-tailed Student's t-test of the respective days independently. *: $p < 0.05$; **: $p < 0.01$; ns, non-significant. A.U., arbitrary units; n/d, not detectable; norm., normalized. (D) Samples from Fig 3G and 3H were analyzed for *IFNL1* and (E) *IFIT1* expression relative to *RPS11* using the $2^{-\Delta Ct}$ method. Data show mean ± SD of n = 4 biological replicates of two independent HLC differentiations. Statistical analysis of ΔORF2 over WT was performed using unpaired two-tailed Student's t-test of each time point independently and is indicated above the respective time points. **: $p < 0.01$; ***: $p < 0.001$; ns, non-significant. A.U., arbitrary units; n/d, not detectable.
(TIFF)

**S10 Fig. Additional scRNA-seq analysis at 56h post-infection.** (A) GSEA was performed, comparing the ΔORF2-infected with the WT-infected sample at 56h post-infection, based on the hallmark gene sets of the Human MSigDB Collections [40]. Positive overlap corresponds to genes upregulated in ΔORF2. (B) UMAP projections of uninfected, WT-, and ΔORF2-infected samples at 56h post-infection, highlighting the number of different genes detected per cell and (C) the number of transcripts detected per cell in all samples. (D) The percentage of infected cells, defined by at least one detected HEV RNA count, was quantified in uninfected, WT-, and ΔORF2-infected samples at 56h post-infection. (E) Normalized HEV RNA counts across infected cells in the inactive and active clusters of WT- and ΔORF2-infected samples at 56h post-infection.
(TIFF)

**S11 Fig. scRNA-seq analysis of WT infection on day 7** (A) UMAP projections of uninfected and (B) WT-infected HepG2/C3A cells, harvested on day 7 post-infection for scRNA-seq analysis. Cells were clustered based on a list of ~400 ISGs published previously by Schoggins *et al.* [41]. (C) Combined UMAP projections of uninfected and WT-infected samples, colored by condition. (D) Combined UMAP projection of the uninfected and WT-infected samples on day 7 post-infection with indicated binarized HEV RNA counts in purple.
(TIFF)

**S1 Table. List of oligonucleotides.**
(DOCX)

## Acknowledgments

The authors gratefully acknowledge Charles Rice, Suzanne Emerson, Koji Ishii, Takaji Wakita, Rainer Ulrich, Frauke Mücksch, and Henrik Kaessmann for sharing reagents, as well as Andrew Freistaedter and Céline Schneider for excellent technical support. We acknowledge Vibor Laketa, head of the Infectious Diseases Imaging Platform (IDIP) at the University Hospital Heidelberg for expert support. The authors thank Marlène Dreux for critical reading of the manuscript. We thank the BIOI2 platform and the I2BC's IT support team for making AlphaFold 2.3 easily accessible at the I2BC.

## Author contributions

**Conceptualization:** Ann-Kathrin Mehnert, Viet Loan Dao Thi.

**Data curation:** Carlos Ramirez Alvarez, Thibault Tubiana, Carl Herrmann.

**Formal analysis:** Ann-Kathrin Mehnert, Sebastian Stegmaier, Carlos Ramirez Alvarez, Jungen Hu, Ana Luisa Costa, Carl Herrmann, Viet Loan Dao Thi.

**Funding acquisition:** Viet Loan Dao Thi.

**Investigation:** Ann-Kathrin Mehnert, Sebastian Stegmaier, Carlos Ramirez Alvarez, Thibault Tubiana.

**Methodology:** Ann-Kathrin Mehnert, Sebastian Stegmaier, Carlos Ramirez Alvarez, Elif Toprak, Vladimir Gonçalves Magalhães, Carla Siebenkotten, Jungen Hu, Ana Luisa Costa.

**Resources:** Daniel Kirrmaier, Michael Knop, Xianfang Wu.

**Software:** Ann-Kathrin Mehnert, Carlos Ramirez Alvarez, Ana Luisa Costa, Thibault Tubiana.

**Supervision:** Carl Herrmann, Marco Binder, Viet Loan Dao Thi.

**Writing – original draft:** Ann-Kathrin Mehnert, Viet Loan Dao Thi.

**Writing – review & editing:** Ann-Kathrin Mehnert, Viet Loan Dao Thi.

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
