## [Decision Letter · Decision Letter 0]

10 Jul 2025

PPATHOGENS-D-25-01443

The hepatitis E virus capsid protein ORF2 counteracts cell-intrinsic antiviral responses to enable persistence in hepatocytes

PLOS Pathogens

Dear Dr. Dao Thi,

Thank you for submitting your manuscript to PLOS Pathogens. After careful consideration, we feel that it has merit but does not fully meet PLOS Pathogens's publication criteria as it currently stands. Therefore, we invite you to submit a revised version of the manuscript that addresses the points raised during the review process.

Please submit your revised manuscript within 60 days Sep 08 2025 11:59PM. If you will need more time than this to complete your revisions, please reply to this message or contact the journal office at plospathogens@plos.org. Please include the following items when submitting your revised manuscript:

We look forward to receiving your revised manuscript.

Kind regards,

Alexander Ploss, Ph.D.

Academic Editor

PLOS Pathogens

Alexander Gorbalenya

Section Editor

PLOS Pathogens

 Sumita Bhaduri-McIntosh

Editor-in-Chief

PLOS Pathogens

orcid.org/0000-0003-2946-9497

 Michael Malim

Editor-in-Chief

PLOS Pathogens

orcid.org/0000-0002-7699-2064

**Additional Editor Comments :**

Thank you for submitting your manuscript for consideration at PLoS Pathogens. Your study has been reviewed by experts in the field and while aspects of your work are certainly interesting a number of concerns were raised that need to be addressed. I would like to draw your attention to concerns raised by by both reviewers with respect to the overall presentation of your data. The plan for addressing points raised during the Reviews Commons review is reasonable.

**Journal Requirements:**

Potential Copyright Issues:

i) Figures 2A, 3A, and 5. We note that the figures are created through BioRender. Please confirm that you hold a Premium account and provide a pdf copy of the CC BY 4.0 Licence as provided by BioRender. For instructions on how to generate a CC BY 4.0 license for your figure, please see the guidelines here: https://help.biorender.com/hc/en-gb/articles/21282341238045-Publishing-in-open-access-resources. 

If you are using the free assets from BioRender, we are unable to publish these images as they are licenced under a stricter licence than CC BY 4.0. In this case we ask you to remove the BioRender images and replace them with open source alternatives.

See these open source resources you may use to replace images / clip-art:

- https://bioart.niaid.nih.gov/ 

- https://bioicons.com/

- https://healthicons.org/ 

- https://scidraw.io/

- https://reactome.org/icon-lib

- https://www.phylopic.org/images 

- https://journals.plos.org/plosbiology/article?id=10.1371/journal.pbio.3002395

5) Thank you for stating "The data is accessible under the GEO accession number GSE288400."  We noticed that the dataset is currently private and is scheduled to be released on January 30, 2029.Please note that, though access restrictions are acceptable now, your entire minimal dataset will need to be made freely accessible if your manuscript is accepted for publication. This policy applies to all data except where public deposition would breach compliance with the protocol approved by your research ethics board. If you are unable to adhere to our open data policy, please kindly revise your statement to explain your reasoning and we will seek the editor's input on an exemption.

2) If any authors received a salary from any of your funders, please state which authors and which funders..

Please ensure that the funders and grant numbers match between the Financial Disclosure field and the Funding Information tab in your submission form. Note that the funders must be provided in the same order in both places as well. Please include the grant numbers in the Funding Information tab.

7) Please provide a completed 'Competing Interests' statement, including any COIs declared by your co-authors. If you have no competing interests to declare, please state "The authors have declared that no competing interests exist". 

**Reviewers' Comments:**

Reviewer's Responses to Questions

**Part I - Summary**

Reviewer #1: (No Response)

Reviewer #2: Mehnert AK et al. are describing in their manuscript the potential role of the ORF2 capsid protein in counteracting the innate and inflammatory responses during HEV infection. The authors employed different cell culture models, i.e. A549, HepG2/C3A, Huh-7.5 and hepatocyte-like cells, together with full-length HEV genomes and infectious viruses. They complemented there investigations by a single cell RNA sequencing analysis of HepG2/C3A cells infected by wt or delORF2 HEV.

The study is taking advantage of original tools (KO cells, mutated full-length genomes) and advanced cell models or techniques (HLC, scRNA-seq) to tackle a very relevant question which has been addressed in the past but with inconsistency in the conclusions.

**Part II – Major Issues: Key Experiments Required for Acceptance**

Reviewer #1: In this study, the authors have provided a comprehensive introduction of previous research on the role of HEV ORF2 and ORF3 in cell-intrinsic antiviral responses. They emphasized the major limitations of these previous studies in particular the use of overexpression approach of viral proteins, and the use of cell line based models. Therefore, the authors aimed to better clarify ORF2 and ORF3 proteins in antiviral response by: 1) use of viral mutants; 2) different immunocompetent hepatocellular systems. However, there are major issues in their claims and just to list a few below.

1. Figure 1, the results are also based on ectopic overexpression of ORF2 and ORF3. By the way, overexpression per se is not a problem, but the authors challenged the previous studies of using this approach while they are still using this method.

2. Figure 1, A549 cell line was used (a human lung carcinoma cell line);

Figure 2, HepG2/C3A cells used (derivative of Hep G2 cell line originally isolated from the liver of a patient with hepatocellular carcinoma);

Figure 3, S10-3 cells (a subclone of the human hepatoma Huh7 cell line) and HepG2/C3A cells, although human pluripotent stem cell-derived hepatocyte-like cells (HLCs) were also used;

Figure 4, HepG2/C3A cells used;

Overall, these models are not fundamentally different or superior to the models used in previous studies, and it is unclear how their models qualified as “immunocompetent hepatocellular systems”. Furthermore, in the title “The hepatitis E virus capsid protein ORF2 counteracts cell-intrinsic antiviral responses to enable persistence in hepatocytes”, the word “hepatocytes” is somewhat deceiving, suggesting the main findings are based on primary hepatocytes or at least stem cell-derived HLCs. In fact, most of the results were obtained from cell lines.

3. Overall, most of the results are descriptive, and even segmented. It is unclear how their findings can further advance the current knowledge in the field.

Reviewer #2: - While the study is very interesting and revised concepts in light of advanced techniques, the current manuscript has somehow difficulty to convey a clear message. The manuscript may greatly gain in readability by splitting some of the figures as well as the results section into distinct paragraphs. In particular, the co-IP experiment showing TBK1 and ORF2 interaction is not well-integrated in Fig 1. The authors may want to reorganize it together with results of Fig 2 in which they show result with TBK1 inhibitor or simply omit this result. Figure 2 may also be divided.

In addition, the text may be streamlined to gain in interest. The authors are extensively justifying their experimental conditions which can be limited. As an example, lines 237-239, the authors explained why BX795 is employed only for 48h, but in my view this can be omitted in the main text as it overloads the results section and dilute the main information.

**Part III – Minor Issues: Editorial and Data Presentation Modifications**

Reviewer #1: (No Response)

Reviewer #2: 1- In Fig1, gene expression is related to GAPDH expression why in Fig 2 and 3 the authors used RPS11 for the normalization. Is there a rationale for this?

2- Title of Fig2 mentions “in immunocompetent cells“. I would remove this notion since panel J has been performed in Huh-7.5 cells.

3- L205-206: “Comparable infection with trans-complemented…“, in the context, the sentence is difficult to understand (trans-complementation system not introduced earlier) and provide complexity. This should be revised.

4- L224-233: the authors are explaining the controls they used in the experiments. It would be better to integrate these informations within the description of the results rather than making this point at the end of the paragraph.

5- Fig 3E, ISG15 is not differentially expressed during infection of HepG2/C3A cells with delORF2 and wt HEV while in scRNA-seq analysis it seems significantly different.

PLOS authors have the option to publish the peer review history of their article (what does this mean? ). If published, this will include your full peer review and any attached files.

**Do you want your identity to be public for this peer review?** For information about this choice, including consent withdrawal, please see our Privacy Policy .

Reviewer #1: No

Reviewer #2: No

**Figure resubmission:**
---

## [Decision Letter · Decision Letter 1]

2 Sep 2025

Dear Dr. Dao Thi,

We are pleased to inform you that your manuscript 'The hepatitis E virus capsid protein ORF2 counteracts cell-intrinsic antiviral responses to enable persistent replication in cell culture' has been provisionally accepted for publication in PLOS Pathogens.

Best regards,

Alexander Ploss, Ph.D.

Academic Editor

PLOS Pathogens

Alexander Gorbalenya

Section Editor

PLOS Pathogens

Sumita Bhaduri-McIntosh

Editor-in-Chief

PLOS Pathogens

orcid.org/0000-0003-2946-9497

Michael Malim

Editor-in-Chief

PLOS Pathogens

orcid.org/0000-0002-7699-2064

Reviewer #1:

Reviewer #2:

Reviewer Comments (if any, and for reference):

Reviewer's Responses to Questions

**Part I - Summary**

Reviewer #1: (No Response)

Reviewer #2: The study by Mehnert et al is now bringing novel information with powerful tools and state-of-the-art techniques in the field.

**Part II – Major Issues: Key Experiments Required for Acceptance**

Reviewer #1: (No Response)

Reviewer #2: no major issues

**Part III – Minor Issues: Editorial and Data Presentation Modifications**

Reviewer #1: (No Response)

Reviewer #2: none

PLOS authors have the option to publish the peer review history of their article (what does this mean? ). If published, this will include your full peer review and any attached files.

**Do you want your identity to be public for this peer review?** For information about this choice, including consent withdrawal, please see our Privacy Policy .

Reviewer #1: No

Reviewer #2: No

---

## [Editor Report · Acceptance letter]

Dear Dr. Dao Thi,

We are delighted to inform you that your manuscript, " 

The hepatitis E virus capsid protein ORF2 counteracts cell-intrinsic antiviral responses to enable persistent replication in cell culture," has been formally accepted for publication in PLOS Pathogens.

Best regards,

Sumita Bhaduri-McIntosh

Editor-in-Chief

PLOS Pathogens

orcid.org/0000-0003-2946-9497

Michael Malim

Editor-in-Chief

PLOS Pathogens

orcid.org/0000-0002-7699-2064